# Establishing a pure antiferroelectric PbZrO$_3$ phase through tensile epitaxial strain

Krina Parmar[1,2], Pauline Dufour[1], Emma Texier[2], Cécile Carrétéro[1], Xiaoyan Li[3], Florian Godel [1], Jirka Hlinka [4], Brahim Dkhil [5], Daniel Sando [6], Hugo Aramberri [7], Jorge Íñiguez-González [7,8], Stéphane Fusil [1], Alexandre Gloter[3], Thomas Maroutian [2] & Vincent Garcia [1] ✉

The nature of lead zirconate, the historical antiferroelectric material, has recently been challenged. In PbZrO$_3$ epitaxial films, thickness reduction engenders competition among antiferroelectric, ferrielectric and ferroelectric phases. All studies so far on PbZrO$_3$ films have utilized commercially-available oxide single crystals with large compressive lattice mismatch, causing the films to undergo strain relaxation. First-principles calculations have predicted that tensile strain can stabilize antiferroelectricity down to the nanometre scale. Here we use tensile strain imposed by artificial substrates of LaLuO$_3$ to stabilize a pure antiferroelectric phase in PbZrO$_3$. Sharp double hysteresis loops of polarization vs electric field show zero remanent polarization, and polar displacement maps reveal the characteristic up-up-down-down antipolar pattern down to 9 nanometre film thicknesses. Moreover, the electron beam can move this antipolar pattern through the nucleation and annihilation of translational boundaries. These results highlight the critical role of coherent epitaxial strain in the phase stability of PbZrO$_3$.

Antiferroelectrics possess a compensated antiparallel arrangement of electric dipoles, resulting in a zero net polarization[1]. Another critical feature is that electric field should convert this antiferroelectric phase into a ferroelectric phase, and this transformation is reversible when turning off the field[2]. This volatile phase transition results in characteristic double hysteresis of polarization as a function of the electric field. As the antiferroelectric-to-ferroelectric phase transition encompasses large variations in charge, volume and temperature, antiferroelectrics are very attractive for applications such as high-density energy storage[3], electromechanical actuators[4], and electrocaloric refrigeration[5].

PbZrO$_3$ was the first material to be considered as antiferroelectric[6]. Still, it remains intensively investigated both

theoretically and experimentally. Below its critical temperature of about 500 K, PbZrO$_3$ transitions between the high-temperature P$m\bar{3}m$ cubic symmetry and the low-temperature P$bam$ orthorhombic symmetry with the characteristic up-up-down-down antipolar pattern of Pb ions. This phase transition is associated with antiferrodistortive oxygen octahedra tilts. Theoretical studies demonstrated the critical role of these tilts in the stabilization of antiferroelectric ordering in PbZrO$_3$[7,8]. Recently, first-principles calculations revealed that another polymorph of PbZrO$_3$ with I$ma2$ space group, showing the same antiferrodistortive tilts but with an uncompensated up-up-down ferrielectric pattern, may actually be the ground state[9]. On the experimental side, scanning transmission electron microscopy (STEM) investigations in PbZrO$_3$ single crystals concluded that both the

---

[1]Laboratoire Albert Fert, CNRS, Thales, Université Paris-Saclay, Palaiseau, France. [2]Centre de Nanosciences et de Nanotechnologies, CNRS, Université Paris-Saclay, Palaiseau, France. [3]Laboratoire de Physique des Solides, CNRS, Université Paris-Saclay, Orsay, France. [4]Institute of Physics, Academy of Sciences of the Czech Republic, Na Slovance 2, Praha 8, Czech Republic. [5]Université Paris-Saclay, CentraleSupélec, CNRS, Laboratoire SPMS, Gif-sur-Yvette, France. [6]MacDiarmid Institute for Advanced Materials and Nanotechnology, School of Physical and Chemical Sciences, University of Canterbury, Christchurch, New Zealand. [7]Luxembourg Institute of Science and Technology (LIST), Avenue des Hauts-Fourneaux 5, Esch/Alzette, Luxembourg. [8]Department of Physics and Materials Science, University of Luxembourg, Rue du Brill 41, Belvaux, Luxembourg. ✉e-mail: vincent.garcia@cnrs-thales.fr

antiferroelectric and ferrielectric phases can coexist[10]. This ferrielectric phase seems to develop via the condensation of translational boundaries[11]. In thick PbZrO₃ films prepared by chemical solution deposition, a stripe-like pattern with alternated antiferroelectric and ferrielectric phases was interpreted as due to residual compressive strain[12]. As the film thickness is reduced in epitaxial PbZrO₃, the situation becomes even more complex. Initial studies showed signatures of ferroelectric phase stabilization via electric measurements[13,14], and recent STEM investigations revealed complex phase transitions from the classical antipolar P*bam* to ferrielectric I*ma*2, orthorhombic or rhombohedral ferroelectric phases, as the thickness of the film is reduced to the 45-5 nm range[15–18]. Indeed, while ferroelectrics tend to break into nanodomains looking like antiferroelectrics when grown as ultrathin films, antiferroelectric PbZrO₃ seems to follow an inverse trend by transiting to a non-zero polar state[19]. Overall, the modern studies on ultrathin films of PbZrO₃ are typically carried out on relaxed layers with possible residual compressive strain. On the other hand, theory suggested that ferroelectricity would be favored under compressive strain, while antiferroelectricity would persist if PbZrO₃ was grown under tensile strain[20].

A long-standing issue for the growth of high-quality epitaxial thin films of PbZrO₃ is the difficulty in finding single-crystal oxide substrates with cell parameters that are well matched with those of PbZrO₃. Indeed, in its bulk antiferroelectric phase, PbZrO₃ displays a P*bam* orthorhombic symmetry with $a_o = 5.882$ Å, $b_o = 11.783$ Å and $c_o = 8.228$ Å[21] (o stands for orthorhombic), resulting in pseudo-cubic parameters of 4.163 Å in the $a_o–b_o$ plane and 4.114 Å along the $c_o$ axis. However, PbZrO₃ epitaxial thin films are usually grown on standard perovskite substrates such as SrTiO₃[12,13,15,17,18,22,23], DyScO₃[4,24] or GdScO₃[25] with unit-cell parameters ranging from 3.905 Å to 3.971 Å, resulting in large compressive strain. This large lattice mismatch causes the PbZrO₃ thin films to relax during growth, forming defects such as edge dislocations[12,14,18]. A related issue is that studying the electrical properties of PbZrO₃, in order to exploit its antiferroelectric properties, becomes challenging when the film thickness is decreased below 50 nm.

In order to reduce the number of defects with epitaxially-strained PbZrO₃ films, we employed a buffer layer of LaLuO₃, a perovskite with large lattice parameters. Single crystals of LaLuO₃ prepared by the Czochralski method were shown to crystallize in a P*bnm* orthorhombic structure[26–28] with $a_o = 5.810$ Å, $b_o = 6.013$ Å, and $c_o = 8.373$ Å[28],

corresponding to a pseudo-cubic lattice parameter of 4.181 Å in the $a_o–b_o$ plane and 4.186 Å along the $c_o$ axis. As there are no single-crystal substrates of LaLuO₃ commercially available, we developed the growth of LaLuO₃ epitaxial thin films as artificial substrates. To our knowledge, Schubert et al. were the only ones to report the growth of epitaxial LaLuO₃ by pulsed laser deposition[29]. Following their work, we were able to stabilize fully-relaxed thick films of LaLuO₃ on commercially-available DyScO₃(110)ₒ single crystals ("Methods", Supplementary Fig. 1). Despite the large lattice mismatch, the LaLuO₃ orthorhombic layer, with typical thicknesses of 100 nm, shows good crystalline quality and grows with the same (110)ₒ orientation as the DyScO₃ substrate. We then used such LaLuO₃-buffered DyScO₃(110)ₒ crystals for the growth of PbZrO₃.

PbZrO₃ thin films with thicknesses ranging from 200 nm down to 9 nm were grown by pulsed laser deposition using a KrF excimer laser. In order to perform electrical measurements, a bottom electrode of SrPbO₃ with typical thicknesses of 15 nm was grown on LaLuO₃ prior to the growth of PbZrO₃ (Methods). SrPbO₃ also crystallizes in a P*bnm* orthorhombic structure with $a_o = 5.852$ Å, $b_o = 5.969$ Å, and $c_o = 8.324$ Å[30,31], corresponding to a pseudo-cubic lattice parameter of 4.179 Å in the $a_o–b_o$ plane and 4.162 Å along the $c_o$ axis. Hence, if the orthorhombic symmetry is preserved and given the lattice constants of SrPbO₃ and LaLuO₃, we expect PbZrO₃ to grow under an anisotropic epitaxial tensile strain of +0.4% and +1.2–1.8% in the two in-plane directions (Supplementary Note 1). Our experimental strategy is motivated by first-principles calculations, which suggested that such tensile strain would stabilize antiferroelectricity[20].

## Results

### Epitaxial arrangement of the films

We first examined the structural properties of these heterostructures using STEM. The cross-section high-angle annular dark field (HAADF) STEM image of the whole epitaxial stack (Fig. 1a) shows well-defined layers with thicknesses of 30 nm, 19 nm, and 108 nm for PbZrO₃, SrPbO₃, and LaLuO₃, respectively. Using local FFTs, we could identify the epitaxial orientation of each layer with respect to their FFT diffraction peaks (color-filtered in Fig. 1b). The blue color of the DyScO₃ substrate and its associated FFT shows that the zone axis of the image is parallel to the $c_o$ axis of the substrate with characteristic ½{110} pseudo-cubic reflections (yellow circles in the FFT of Fig. 1b) corresponding to the $a_o$ and $b_o$ orthorhombic axes and the (110)ₒ surface

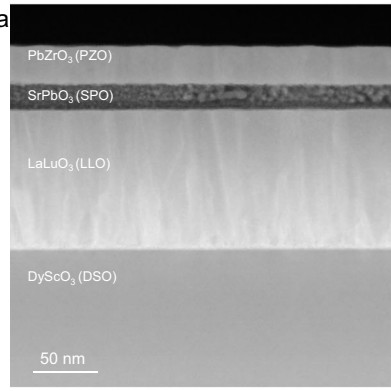
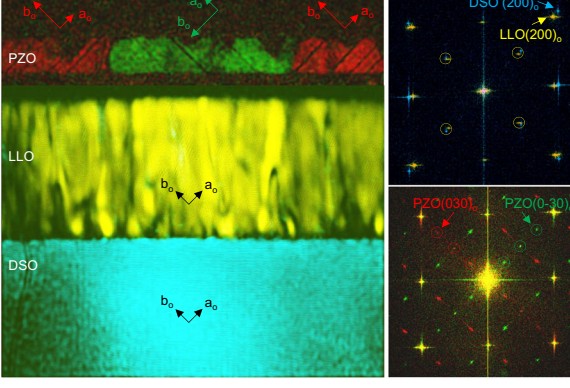
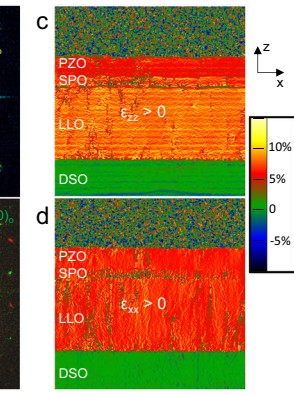

**Fig. 1 | Structural analysis of the complex epitaxial stack using scanning transmission electron microscopy. a** HAADF-STEM image of the carbon-covered PbZrO₃/SrPbO₃/LaLuO₃ orthorhombic perovskite layers grown on the DyScO₃(110)ₒ orthorhombic substrate. The layer thicknesses are 30, 19, and 108 nm for PbZrO₃, SrPbO₃, and LaLuO₃, respectively. The SrPbO₃ electrode reacts under the electron beam of the microscope. **b** Colored images obtained by filtering the FFT contributions of the different layers (as displayed on the right panel), showing DyScO₃ (DSO), LaLuO₃ (LLO), and the two orientations of PbZrO₃ (PZO) corresponding to 90° rotation around the $c_o$ axis. The ½{110} superlattice reflections of DyScO₃ and

LaLuO₃ are emphasized by the yellow circles. The ¼{110} superlattice reflections of PbZrO₃ are underlined by the red and green circles. The zone axis is parallel to the $c_o$ axes of the four orthorhombic layers, giving rise to the following epitaxial relationship: PbZrO₃(120)ₒ,(1-20)ₒ ‖ SrPbO₃(110)ₒ ‖ LaLuO₃(110)ₒ ‖ DyScO₃(110)ₒ with PbZrO₃[001]ₒ ‖ SrPbO₃[001]ₒ ‖ LaLuO₃[001]ₒ ‖ DyScO₃[001]ₒ. **c, d** Out-of-plane unit-cell deformation ($\varepsilon_{zz}$) (**c**) and in-plane unit-cell deformation ($\varepsilon_{xx}$) (**d**) obtained from geometrical phase analysis of the bright field STEM image (Supplementary Fig. 4). The uniform lattice expansion observed for the in-plane unit-cell deformation shows that the epitaxial stack is coherently strained to the LaLuO₃ buffer layer.

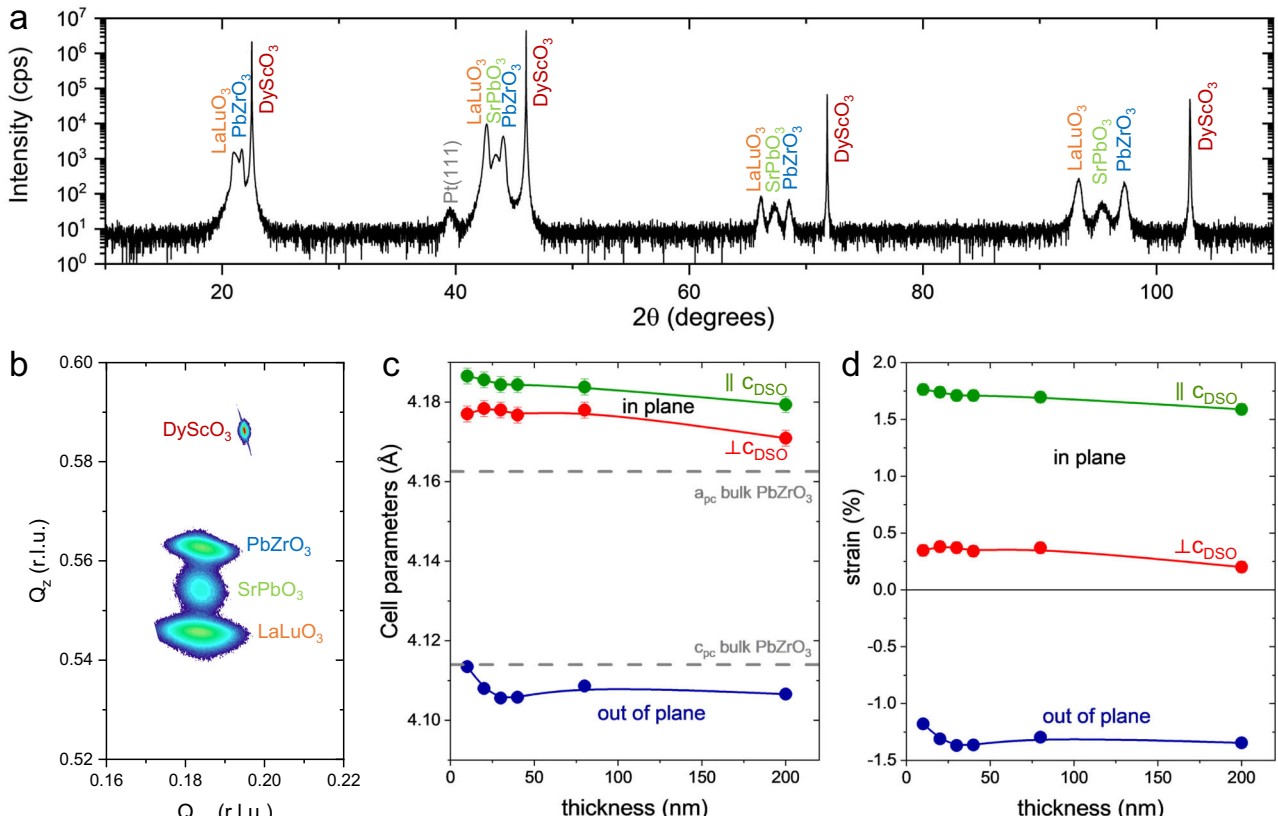

**Fig. 2 | Epitaxial PbZrO₃ thin films under tensile strain. a** $2\theta$-$\omega$ X-ray diffraction pattern showing the $(hh0)_o$ symmetric peaks of the DyScO₃, LaLuO₃ and SrPbO₃ as well as the $(h,2h,0)_o$ peak for PbZrO₃, with $h = 1, 2, 3, 4$ for layer thicknesses of 30, 19, and 108 nm for PbZrO₃, SrPbO₃, and LaLuO₃, respectively. The small peak at around 40° corresponds to the Pt(111) of the capacitors. **b** Reciprocal space map around DyScO₃(332)ₒ for a similar sample as in (**a**) with a PbZrO₃ thickness of 40 nm. The three epitaxial layers of LaLuO₃, SrPbO₃, PbZrO₃ share a common in-plane lattice parameter. **c** Pseudo-cubic in-plane (green and red) and out-of-plane (blue) lattice parameters of PbZrO₃ as a function of the film thickness estimated from reciprocal space maps around DyScO₃(220)ₒ, (420)ₒ, (332)ₒ, (240)ₒ, (33-2)ₒ (Supplementary Fig. 10 and Supplementary Fig. 11, Supplementary Note 1). The bulk parameters $a_{pc}$ and $c_{pc}$ of PbZrO₃ are shown as dashed lines. **d** Epitaxial strain for the two in-plane axes and the out-of-plane axis deduced from c (Supplementary Note 1). A large in-plane tensile strain is measured along the PbZrO₃ $c_o$ axis (green) and a more moderate one perpendicularly to the $c_o$ axis (red). This results in a strong out-of-plane compressive strain (blue). Overall, the tensile epitaxial strain does not vary significantly with the PbZrO₃ film thickness from 9 to 200 nm. The error bars in (**c**) and (**d**) represent the standard deviation; if not visible, they are smaller than the symbol size.

orientation. As previously mentioned, the orthorhombic layer of LaLuO₃ follows the same $(110)_o$ orientation with an overall homothety of the unit-cell (see the FFT with yellow color). SrPbO₃ appears sensitive to either the sample thinning process or the electron beam, and its crystallinity is not preserved throughout the whole cross-section specimen. Nevertheless, local observations confirm that SrPbO₃ grows with the same $(110)_o$ orientation as the LaLuO₃ and DyScO₃ (Supplementary Fig. 2). Hence, the SrPbO₃/LaLuO₃/DyScO₃ stack harbors a single crystal-like $(110)_o$ orientation with all the $c_o$ axes parallel with one another. Regarding the PbZrO₃ film, FFTs show two diffraction patterns (green and red) with ¼{110} superlattice reflections (green and red circles in the FFT of Fig. 1b). Indeed, the antiferroelectric P*bam* phase of PbZrO₃ consists of antiparallel up-up-down-down Pb displacements, forming stripes with a periodicity of four layers in the {110} pseudo-cubic plane, often coined as commensurate modulations[32] of ¼{110}. Consequently, these two FFTs correspond to a $c_o$ axis of the film parallel to the zone axis and with $(120)_o$ and $(1\bar{2}0)_o$ orientations, as sketched in Fig. 1b. The antiferroelectric domains can extend over typical lateral sizes of 150 nm, significantly larger than the 30-nm PbZrO₃ thickness. Thus, the number of possible variants is reduced from six[25] possible to only two, due to the preserved orthorhombic symmetry throughout the epitaxial stack. The presence of dark lines in the PbZrO₃ domains (Fig. 1b) is attributed to antiphase boundaries, but the film remains purely in the antipolar P*bam* phase (Supplementary Fig. 3). From the geometrical phase analysis (Fig. 1c, d and

Supplementary Fig. 4), we observe that the large lattice mismatch strain (>5%) between LaLuO₃ and DyScO₃ is relieved within the first 10 nm of the layer. Additional strain profile analyses taken on different areas with weakly (Supplementary Fig. 5) or strongly amorphized SrPbO₃ layers (Supplementary Fig. 6) show that LaLuO₃, SrPbO₃, and PbZrO₃ display a constant in-plane lattice parameter value, suggesting that the epitaxial stack is coherently strained to the LaLuO₃ buffer layer.

## Evaluation of the epitaxial strain
We used X-ray diffraction to get more quantitative insights into the structure of the epitaxial heterostructure. A typical $2\theta$-$\omega$ X-ray diffraction pattern is displayed in Fig. 2a, showing high crystalline quality for the LaLuO₃, SrPbO₃ and PbZrO₃ thin films and no secondary phases (other film thicknesses are displayed in Supplementary Fig. 7). Wide range reciprocal space maps taken with a 2D detector (Supplementary Fig. 8) confirm the overall epitaxy with PbZrO₃(120)ₒ ∥ SrPbO₃(110)ₒ ∥ LaLuO₃(110)ₒ ∥ DyScO₃(110)ₒ and PbZrO₃[001]ₒ ∥ SrPbO₃[001]ₒ ∥ LaLuO₃[001]ₒ ∥ DyScO₃[001]ₒ. The asymmetric high-resolution reciprocal space map around DyScO₃(332)ₒ (Fig. 2b) shows that LaLuO₃, SrPbO₃ and PbZrO₃ share a common in-plane lattice constant, and this is the case for all thicknesses between 200 nm and 9 nm (Supplementary Fig. 9). We also performed reciprocal space maps around the (420)ₒ, (332)ₒ, (240)ₒ, and (33$\bar{2}$)ₒ of DyScO₃ (Supplementary Fig. 10 and Supplementary Fig. 11) for the 200, 80, 40, 30, 20, and 9 nm

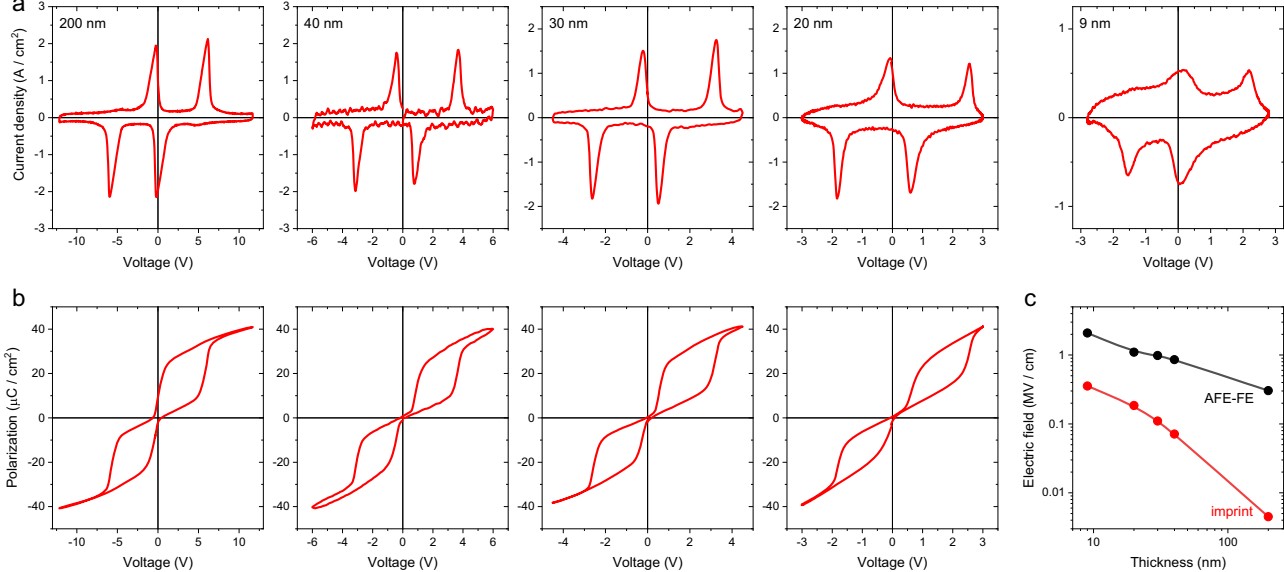

**Fig. 3 | Thickness dependence of the electrical properties of Pt/PbZrO₃/SrPbO₃ capacitors grown on LaLuO₃-buffered DyScO₃ substrates. a** Current vs. voltage loops for a triangular waveform of voltage at 2 kHz for film thicknesses of 200, 40, 30, 20, and 9 nm (from left to right). Four displacement current peaks are observed for all the film thicknesses. **b** Corresponding polarization vs. voltage loops show clear double hysteresis, characteristic of antiferroelectric capacitors. Capacitive contributions for the lowest film thickness of 9 nm impede the integration of polarization vs voltage. **c** Critical electric field of the antiferroelectric-to-ferroelectric (AFE-FE) phase transition as a function of the film thickness (black, in log-log scale). The AFE-to-FE critical electric field is calculated by averaging the positive and negative critical voltages in (**b**), and dividing this average voltage by the film thickness. The imprint field, corresponding to the shift of the loops towards positive voltage, is also plotted (in red) as a function of the PbZrO₃ thickness. Voltage is applied to the SrPbO₃ electrode while the Pt top electrode is grounded.

thicknesses of PbZrO₃ to calculate the lattice parameters of PbZrO₃ (Fig. 2c). For all film thicknesses, the in-plane pseudo-cubic lattice parameters along $c_o$ (Fig. 2c, green dots) and perpendicular to $c_o$ (Fig. 2c, red dots) are larger than the bulk parameters (Fig. 2c, dashed gray lines). By contrast, the out-of-plane parameters (Fig. 2c, blue dots) are smaller than the bulk values, in line with a biaxial epitaxial tensile strain. Considering this (120)ₒ epitaxy of PbZrO₃, we estimated the strain for the two in-plane and the out-of-plane directions (Fig. 2d and Supplementary Note 1). The films display a relatively constant anisotropic in-plane tensile strain as a function of thickness, reaching +1.7 ± 0.1% along $c_o$ and +0.3 ± 0.1% perpendicular to $c_o$, in agreement with fully-strained layers on the artificial substrate of LaLuO₃. The out-of-plane strain is estimated to be −1.3 ± 0.1 %. Hence, growing PbZrO₃ on SrPbO₃/LaLuO₃-buffered DyScO₃ substrates, we were able to successfully impose a coherent tensile strain that is robust over a wide range of film thicknesses.

## Electrical properties of the films

We now explore the influence of such an epitaxial tensile strain on the antiferroelectric properties of PbZrO₃. Capacitors were fabricated on the PbZrO₃/SrPbO₃/LaLuO₃ stack by a combination of optical lithography and lift-off of sputtered Pt top square electrodes, with lateral sizes varying from 5 to 120 μm. The electrical properties of typical 30 × 30 μm² capacitors are displayed in Fig. 3 for PbZrO₃ thicknesses of 200, 40, 30, 20, and 9 nm, under a triangular voltage excitation with a frequency of 2 kHz. Current vs. voltage loops (Fig. 3a) for thicknesses of 20-200 nm show four sharp current peaks characteristic of antiferroelectric-to-ferroelectric field-induced phase transitions. By integrating the displacement current densities as a function of time, we obtain the corresponding polarization vs. voltage (Fig. 3b), which shows clear double hysteresis loops. Despite the low 20–40 nm film thickness, no remanent polarization is detected, with no evidence of ferroelectric phases, attesting to the high quality of these epitaxial PbZrO₃ films. Remarkably, these displacement current peaks are still clearly detectable for 9-nm-thick PbZrO₃ films, though a large

capacitive background impedes reliable polarization integration. The critical fields for the antiferroelectric-to-ferroelectric phase transitions increase from 0.3 MV/cm to 2.4 MV/cm when the film thicknesses decrease from 200 nm to 9 nm (Fig. 3c). Such large critical fields are consistent with measurements in pure PbZrO₃ single crystals[33], while incommensurate phases stabilized by doping these crystals show reduced critical fields[32]. A possible scenario is that the ferroelectric phase needs to nucleate first in a pure antiferroelectric matrix, while local ferrielectric or ferroelectric coexisting phases act as nucleation points in doped PbZrO₃. Another striking feature of these double hysteresis loops is the horizontal voltage shift that tends to increase as thickness decreases (Fig. 3b). This imprint field, reaching 0.3 MV/cm for the lowest PbZrO₃ thickness (Fig. 3c), could be related to the asymmetry between the Pt/PbZrO₃ and PbZrO₃/SrPbO₃ interfaces. Overall, the capacitors based on epitaxial PbZrO₃ grown on SrPbO₃/LaLuO₃ display exceptional electrical properties with sharp double hysteresis loops down to 20 nm and antiferroelectric signatures down to 9 nm thicknesses.

## Visualization of the atomic-scale polar displacements

To confirm that this electrical behavior is associated with the antiferroelectric nature of PbZrO₃, we used high-resolution HAADF-STEM imaging to map the local polar displacements of the Pb atoms (Fig. 4 and Supplementary Fig. 12) for the 30, 20, and 9 nm thick films. The 30 nm PbZrO₃ film displays stripes with polar displacements alternating along <110> pseudo-cubic directions for every two {110} planes (inset of Fig. 4a). This characteristic up-up-down-down antipolar pattern is expected for the antiferroelectric phase of PbZrO₃ with P*bam* symmetry. Two antipolar domains are observed in Fig. 4a with a 90-degree domain wall aligned along the {100} pseudo-cubic plane. The dipoles locally form head-to-tail boundaries in order to minimize the depolarization energy, as also observed for PbZrO₃ ceramics[32]. Interestingly, this characteristic antipolar pattern is preserved for PbZrO₃ film thicknesses of 20 nm (Fig. 4b) and 9 nm (Fig. 4c), though the interfaces may introduce some local polar disorder (e.g., bottom right

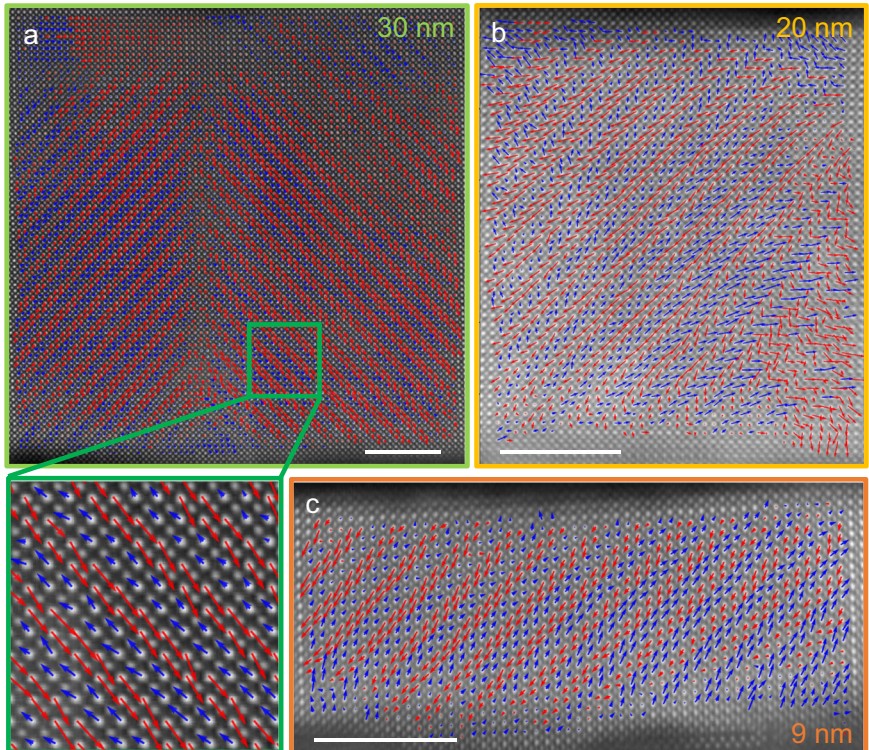

**Fig. 4 | Antipolar texture of PbZrO₃ thin films using scanning transmission electron microscopy.** Polar displacements were obtained by comparing the location of the Pb with respect to the barycenter of the Zr square lattice, in high-resolution HAADF-STEM images (raw images are displayed in Supplementary Fig. 12). In all the images, the zone axis is parallel to the $c_o$ axis of DyScO₃. The resulting dipoles are represented as colored arrows. **a** Polar textures of the 30-nm-thick PbZrO₃ film, showing two large antipolar domains with a 90-degree domain wall. The bottom inset is a zoom emphasizing the characteristic up-up-down-down antipolar pattern of the P*bam* phase. **b** Polar textures in the 20-nm-thick PbZrO₃ film. **c** Polar textures in the 9-nm-thick PbZrO₃ film. Horizontal scale bars are 5 nm.

corner of Fig. 4b). The antipolar nature of these 30 nm to 9 nm-thick PbZrO₃ films is strikingly different from multiple recent results on relaxed epitaxial thin films of PbZrO₃ with similar thicknesses, reporting homogeneous polar displacements characteristic of a ferroelectric phase[15,17,18]. Overall, we have demonstrated that the excellent electrical properties of the PbZrO₃ films grown under tensile epitaxial strain are associated with their true antipolar nature down to 9 nm.

Regarding this thinnest PbZrO₃ film, while a clear antipolar pattern is observed in its pristine state, we discovered instabilities of this pattern under the STEM electron beam. In PbZrO₃ single crystals, similar time-dependent STEM experiments revealed a transition from the antipolar to a cycloidal polar state under the electron beam[34]. In our case, these instabilities were not detected in thicker PbZrO₃ films. Sequential HAADF-STEM images taken every two seconds allowed us to visualize changes in the antipolar pattern (Fig. 5). Indeed, while the up-up-down-down antipolar pattern is initially very well ordered over the whole region of interest (Fig. 5a, *t* = 0–38 s), local defects start to nucleate after several scans (Fig. 5a, *t* = 40 s). We first observe the nucleation of −π/2 translational boundaries[10,11] characterized by a phase shift of the up-up-down-down square (4 unit-cell) wave by 3 unit cells (Fig. 5b, yellow). The nucleation of such a topological defect gives rise to a local uncompensated up-up-down-down-down pattern. This local defect promotes the subsequent shift of the whole antipolar pattern from left to right by one unit cell (Fig. 5b, *t* = 42 s). This cascade-like movement is enabled by the nucleation and annihilation of +π/2 (phase shift by 1 unit cell) translational boundaries[10,11] in other areas, resulting in transient up-down-down (Fig. 5b, black) polar patterns. Consequently, the instability of the antipolar order under the STEM electron beam suggests competing ferrielectric ordering for such low PbZrO₃ thicknesses. Nevertheless, we did not detect the aggregation of translational boundaries in the form of ferrielectric phases[10,12,17] throughout

the different areas of the cross-section specimen. Further investigations are required to identify the possible roles of the interfaces and surfaces in the nucleation of the translation boundaries.

## Discussion

To summarize, we have fabricated pure PbZrO₃ antiferroelectric thin films by pulsed laser deposition. The thin films were successfully grown under tensile strain on LaLuO₃ artificial substrates and did not show any strain relaxation from 9 to 200 nm. Local atomic map investigations by scanning transmission electron microscopy show the characteristic up-up-down-down dipole pattern of the orthorhombic P*bam* phase, with no indication for ferroelectric or ferrielectric phase coexistence, down to 9 nm. Capacitors based on these PbZrO₃ films display sharp double hysteresis of polarization as a function of the electric field with no remanent polarization, evidencing purely antiferroelectric behavior. Our results give clear evidence that the proposed epitaxy stabilizes the antiferroelectric phase against competing ferroelectric or ferrielectric phases down to thicknesses of 9 nm. These findings suggest that epitaxial tensile strain can stabilize antiferroelectricity in PbZrO₃ down to nanometer thicknesses, opening the way for antiferroelectric-based nanodevices.

## Methods

### Sample preparation

All the films were grown by pulsed laser deposition using a KrF excimer laser (Coherent Compex Pro 110, 248 nm wavelength), with sintered ceramic targets (Toshima Manufacturing Co.) of stoichiometric composition for LaLuO₃ and SrPbO₃, and with 10% Pb excess for PbZrO₃. The DyScO₃ substrates were preliminarily annealed at 1000 °C for 3 h under pure O₂ flow, and single termination was systematically checked with atomic force microscopy (Bruker Innova). The 100 nm-thick LaLuO₃

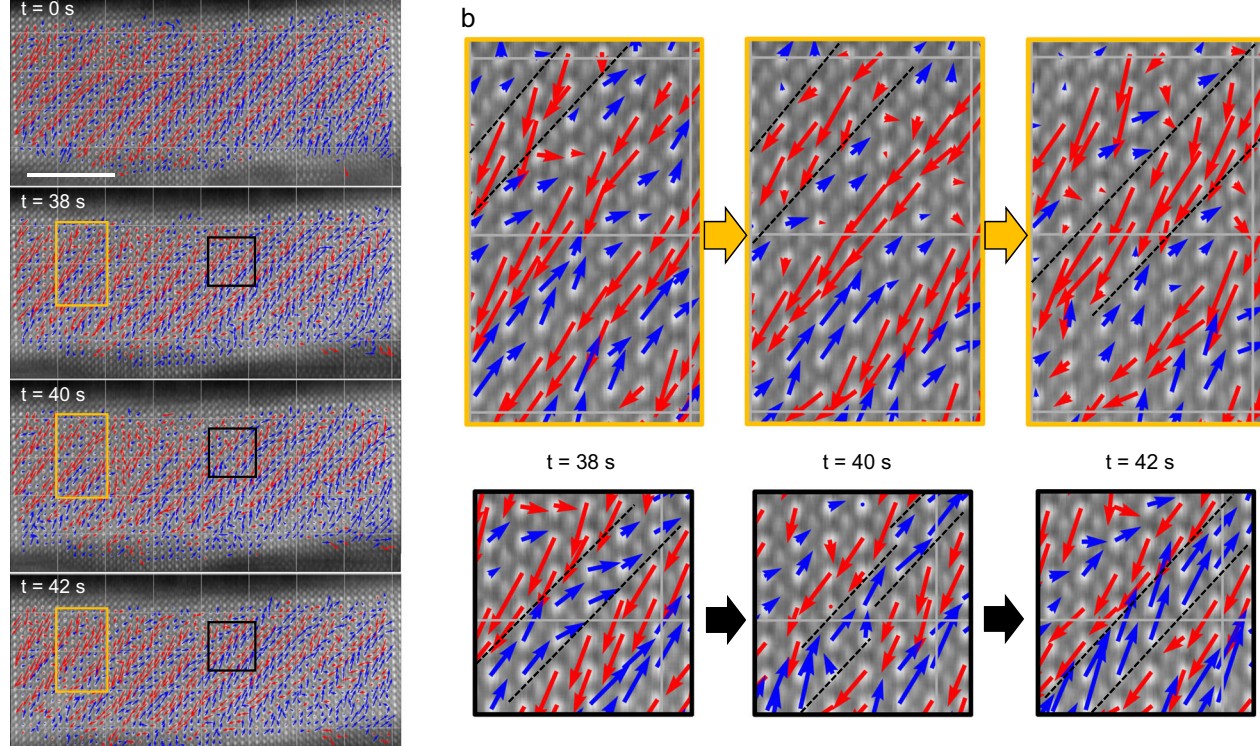

**Fig. 5 | Antipolar instabilities in the 9-nm-thick PbZrO₃ film: translation boundary nucleation and movement of the antipolar pattern under the electron beam. a** Sequential STEM-HAADF images taken with 2 s integration time on the same area, displayed at 0, 38, 40 and 42 s. **b** Zooms taken from the left images for the regions in yellow and black, emphasizing local modification of the antipolar pattern. The horizontal scale bar is 5 nm.

layer was grown at 730 °C under a dynamic O₂ pressure of 0.4 mTorr, at 5 Hz laser repetition rate and 2.5 J/cm² fluence on the target. The sample temperature was then reduced to 560 °C, while O₂ was replaced by N₂O with dynamic pressures of 40 mTorr and 120 mTorr for SrPbO₃ and PbZrO₃, respectively. For both materials, higher growth temperatures result in high lead deficiency and increased surface roughness. The 15 nm-thick SrPbO₃ layer was grown at a laser repetition rate of 2 Hz and 1 J/cm² fluence, and these values were then increased to 5 Hz and 2.5 J/cm² for the PbZrO₃ layer. Lower fluence gives rise to non-stoichiometric precipitates on the film surface. Following the growth, the sample was cooled down to room temperature under 300 Torr of static O₂ pressure. The top electrodes, composed of 20 nm of Pt, were deposited by RF magnetron sputtering.

#### Electrical measurements
P-E hysteresis loops were recorded at room temperature with a Radiant Multiferroic tester.

#### X-ray diffraction
Conventional 2θ-ω scans and reciprocal space maps were performed using Cu K$_{\alpha-1}$ radiation in a 9-kW rotating anode Rigaku SmartLab diffractometer.

#### Scanning transmission electron microscopy
STEM images were acquired using a Cs-corrected USTEM Nion microscope at 100 and 200 keV. Polar displacements were evaluated using two-dimensional Gaussian fitting of the atomic positions in the HAADF images, as implemented in the Atomap software[35]. The off-center displacements of Pb cations were calculated with respect to the geometric center of the four surrounding Zr-site columns, and the arrows representing the off-center intensities and directions were overlaid on the HAADF images at the Pb column positions.

### Data availability

The data that support this work are available via Zenodo at https://doi.org/10.5281/zenodo.15721666.

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

## Acknowledgements

This work is supported by a France 2030 government grant managed by the French National Research Agency (ANR-22-PEEL-0010). We acknowledge support from the European Union's Horizon 2020 research and innovation program under grant agreement no. 964931 (TSAR). We thank the French national network RENATECH for nanofabrication. Work at LIST funded by the Luxembourg National Research Fund (FNR) through Grant C21/MS/15799044/FERRODYNAMICS.

## Author contributions

V.G. proposed the experiments and coordinated the project. K.P., P.D., and E.T. grew the epitaxial thin films under the supervision of T.M. K.P., P.D., C.C., and V.G. performed the X-ray diffraction experiments and analyzed the structure of the films with the help of D.S. A.G. and X.L. prepared the cross-section specimen by focused ion beam, examined the samples by scanning transmission electron microscopy, and calculated the polar displacements. K.P., F.G., P.D., and E.T. fabricated the devices and measured their electrical properties under the supervision of S.F. All the authors, including B.D., J.H., H.A., and J.I.-G., participated in the analysis, the discussion and interpretation of the results. V.G. wrote the manuscript with inputs from all the authors.

## Competing interests

The authors declare no competing interests.
