## [Transparent Peer Review file · Nature Communications]

Establishing a pure antiferroelectric PbZrO₃ phase through tensile epitaxial strain

Corresponding Author: Dr Vincent Garcia

Version 0:

Reviewer comments:

Reviewer #1

(Remarks to the Author)

Reviewer

Advise: Major Revision

This study successfully stabilized a pure antiferroelectric phase in ultrathin PbZrO₃ films by introducing a LaLuO₃ buffer layer to impose tensile epitaxial strain. A systematic investigation of the structural and electrical properties, as well as the electron beam-induced dynamic behavior of polar patterns, was conducted. Utilizing STEM, XRD, and electrical measurements, the study achieves a significant breakthrough in maintaining antiferroelectricity even at a thickness of 9 nm, underscoring its considerable scientific and application potential. While the experimental design is rigorous and well-supported by data, further refinement is needed in the theoretical mechanism analysis and comparative discussion with existing literature.

- 1.The manuscript lacks an in-depth analysis of the potential impact of the LaLuO₃ buffer layer on interfacial defects and the long-term stability of lattice mismatch. It is recommended to supplement the study with high-resolution STEM analysis of the interface structure or theoretical calculations, such as interfacial energy and stress distribution.
- 2.The experiment revealed a significant increase in the critical electric field as the film thickness decreased (from 0.3 MV/cm to 2.4 MV/cm). However, the study merely hypothesizes that "the nucleation of the ferroelectric phase is hindered in the pure antiferroelectric matrix." It is recommended to incorporate phase-field simulations or first-principles calculations to quantitatively analyze the effect of thickness on the phase transition energy barrier.
- 3.Although the study cites theoretical predictions (e.g., Reference 20) supporting the stabilization of the antiferroelectric phase under tensile strain, it does not incorporate computational data (e.g., strain-phase diagrams) to quantitatively validate the experimental results.
- 4.Compared to recent similar studies (e.g., References 17 and 18, which report the coexistence of ferroelectric phases under compressive strain), a more detailed discussion is needed on the physical mechanisms through which strain direction influences the stabilization of ferroelectric and antiferroelectric phases in thin films.
- 5.The image annotations in the manuscript should be clearer (e.g., STEM labeling in Figure 1). Additionally, the Methods section should provide a more detailed description of certain parameters, such as the influence of laser energy density on film crystallinity, to enhance experimental reproducibility.
- 6.According to the reference (Phys. Rev. Lett. 2015, 115, 097601), the critical thickness for AFE PZO thin films is about 6.5 nm. Whether a thickness below 9 nm can remain stay a AFE structure?
- 7.In Figure 1, I cannot found the thickness information about this picture. The author should describes it in the STEM images. A 9-nm-thickness HAADF-STEM image may be more valuable than other thicknesses.
- 8.In Figure 2a, XRD data only show one thickness, but the thickness was not given, and other thicknesses were lacking. Moreover, the 2θ XRD pattern that starts from 38 to 50 degree is enough.
- 9.In Figure 3b, the P-E loops of 9-nm-thickness is lacking, even through the author explain that curves cannot be obtained due to a capacitive contributions.
- 10.In page 3 line 98, there are some unfortold description that (we expect PbZrO₃ to grow under an anisotropic epitaxial tensile strain of +0.4% and +1.2-1.8% in the two in-plane directions), did the special tensile value is suitable for the stability of AFE phase?
- 11.In page 4 line 164, (Such large critical fields are consistent with measurements in pure PbZrO₃ single crystals) should provide a new reference to support the statement.
- 12.The supplementary was not mentioned in the manuscript, the author may explain it at proper position.
13. In the bottom inset of Figure 4a, the altitude of the red antipolar pattern is quite higher than the blue one. Does these

structures are uncompensated or the fitting algorithm is accurate enough? The author can explain this.

14. In Figure 5, why the antipolar pattern is not stable in the 9-nm-thick PbZrO₃ film or it is general phenomena? A theoretical word to explain the dynamics is also lacking.

Reviewer #2

(Remarks to the Author)

This manuscript presents an elegant and well-executed study on the stabilization of a pure antiferroelectric phase in ultrathin PbZrO₃ films under tensile epitaxial strain. The authors convincingly demonstrate that careful strain engineering using LaLuO₃-buffered DyScO₃ substrates allows the realization of an antipolar ground state even in films as thin as 9 nm. The combination of high-quality thin film synthesis, detailed structural characterization using scanning transmission electron microscopy (STEM), and electrical measurements makes a compelling case for the emergence of antipolar instabilities as a function of thickness. The work is timely and addresses a central challenge in the field—namely, the stabilization of nonpolar or antipolar states at reduced dimensions, which is of great interest for antiferroelectric-based applications. Moreover, the observation of distinct dipolar patterns, and their evolution under electron beam irradiation, opens up new questions regarding the interplay between external perturbations and intrinsic lattice instabilities. Overall, I believe this study makes a meaningful contribution to the ongoing effort to understand and control ferroic orders in reduced dimensions and complex oxide heterostructures. However, I suggest the authors address the following points to further improve the clarity, completeness, and rigor of the manuscript:

1. Clarification of strain relaxation in the buffer layer stack: The authors use SrPbO₃/LaLuO₃-buffered DyScO₃ substrates to impose tensile strain in PbZrO₃ films. Given the crucial role of strain in stabilizing the antiferroelectric phase, it would be helpful if the authors could elaborate on how strain evolves across the DyScO₃ substrate, LaLuO₃, and SrPbO₃ layers. In particular, considering that both LaLuO₃ and SrPbO₃ serve as tensile buffer layers, and that SrPbO₃ is known to be vulnerable to electron beam exposure, further clarification is needed on the specific function and necessity of including SrPbO₃, rather than simply using LaLuO₃, in the buffer stack.
2. STEM characterization of the PbZrO₃/SrPbO₃ interface: Despite the e-beam sensitivity of SrPbO₃, high-resolution STEM images of the PbZrO₃/SrPbO₃ interface—either immediately post-FIB or after TEM sample preparation—would be highly informative. Such data would help clarify the microstructural integrity and strain transfer across the interface, and allow readers to better assess the role and effectiveness of the tensile strain buffer layers.
3. Visualization of polar configurations in HAADF-STEM: In Figures 4 and 5, the up-up-down-down and other polar configurations are illustrated using numerous blue and red arrows. While this visual aid is helpful in identifying local dipole arrangements, the use of extensive artificial annotations—without a clear explanation of how the dipole directions are extracted—may be misleading for readers unfamiliar with the technique. The authors are encouraged to explicitly describe the methodology used for determining dipole orientations (e.g., Atomap, Gaussian fitting, or other image processing tools), and to provide side-by-side versions of the images with and without the arrow overlays for better comparison and interpretation.
4. Reversibility of electron beam-induced changes: The electron beam-induced transformation of the antipolar texture in the 9-nm-thick PbZrO₃ film is fascinating. However, the reversibility of this phenomenon remains unclear. Can the authors clarify whether the antipolar pattern reappears after the electron beam is turned off for a sufficient period? This would help determine whether the beam-induced changes are reversible or represent a permanent structural transition. A brief discussion of this point in the main text would be helpful.

In summary, this is a carefully conducted and high-quality study that advances our understanding of antiferroelectricity in strained oxide films. I recommend minor revisions to address the points above, which would further strengthen the manuscript's clarity and impact.

Reviewer #3

(Remarks to the Author)

The polar structures and final responses of PbZrO₃ (PZO) under different external fields are classic issues in the antiferroelectric field. Since the lattice parameters of PZO are significantly larger than those of commonly used perovskite oxide single crystal substrates, it is difficult to obtain coherently strained PZO films. In this manuscript, the authors have grown LaLuO₃ (LLO) layer on the DyScO₃ (DSO) substrate, which is almost fully relaxed thus the large lattice of LLO was preserved. And they claim that the PZO films can be coherently strained on this buffer layer, thus a “pure” antiferroelectric PZO phase can be obtained.

While the work is successful in obtaining LLO as a buffer layer with large lattice parameters, the results regarding the PZO films are not sufficiently remarkable. It is worth noting that the approach of using a buffer layer to obtain high-quality PZO films or the pure-phase PZO has been reported in previous studies, as cited in References 4 and 23. Moreover, these earlier works not only achieved the ideal antiferroelectric PZO films but also utilized PZO in high-performance thermal switching chips or electromechanical devices. Therefore, the novelty of the current work is insufficient for publication in Nature communications.

Technically, there are some obvious details seem indicating that the phases here in PZO are not “pure”:

1. In Figure 1b, distinct strip domains are visible within the PZO layer, yet the authors have not provided any discussion on these features. The presence of these domains implies that the PZO films are not in a pure state.
2. Since the LLO layer is thicker, and is fully relaxed, there should be lots of threading dislocations inside the LLO layer. Moreover, these threading dislocations should be reproduced no matter what layer was grown on it (when coherent along

the in-plane direction). Where are these threading dislocations? Why they seem doesn't affect the responses of PZO? Note that the mismatch here for LLO and DSO is large, which should produce lots of threading dislocations connecting with the misfit dislocations (as illustrated in Fig. S1).

3. The authors state that the GPA strain analysis was performed on the "bright field STEM image". However, i don't see any "bright field STEM image" in Fig. 1.

4. Although the authors say that the SPO was sensitive to electron beam, I notice that in the strain maps of Fig. 1c and 1d, most parts of the SPO layer were still OK. Therefore, it is crucial for the authors to present magnified images that clearly show the coherent PZO/SPO interface. It is reasonable to assume that such an interface exists; it simply needs to be visually presented through magnified views.

Version 1:

Reviewer comments:

Reviewer #1

(Remarks to the Author)

The authors have addressed my concerns in the revisions, and I have no competing financial or non-financial interest in relation to this manuscript , I recommend this paper accept for publication in NC.

Reviewer #2

(Remarks to the Author)

The authors have conducted additional experiments and provided further clarification along with an improved scientific discussion. In light of these revisions, I would recommend the manuscript for publication in Nature Communications.

Reviewer #3

(Remarks to the Author)

The authors have adequately addressed the majority of the concerns from the reviewers; however, two critical issues still require further clarification in the following aspects:

First, PbZrO_3 (PZO) is intrinsically antiferroelectric in its bulk form, the authors tend to claim that in-plane tensile strain stabilizes the antiferroelectric phase. However, this assertion lacks comparative evidence. To substantiate the strain-induced enhancement, quantitative comparisons with unstrained or bulk PZO are essential—for instance, through metrics such as polarization magnitude, ordering temperature, or the critical thickness, etc. Clarify this issue will enhance the novelty the work.

Second, the strain values represented in Fig. 2d, which I believe have been seriously exaggerated. A rough estimation gives the in-plane strain of +0.4% (4.181-4.163/4.163), and the out-of-plane strain of -0.2% (4.105-4.114/4.114). These values are much smaller than that presented in Figure 2d (~ +1.7% and -1.4%).

Version 2:

Reviewer comments:

Reviewer #3

(Remarks to the Author)

The authors have provided further clarifications to my concerns, strengthening the novelty of the manuscript. I recommend acceptance in Nature communications.

Reviewer #1 (Remarks to the Author):

This study successfully stabilized a pure antiferroelectric phase in ultrathin PbZrO₃ films by introducing a LaLuO₃ buffer layer to impose tensile epitaxial strain. A systematic investigation of the structural and electrical properties, as well as the electron beam-induced dynamic behavior of polar patterns, was conducted. Utilizing STEM, XRD, and electrical measurements, the study achieves a significant breakthrough in maintaining antiferroelectricity even at a thickness of 9 nm, underscoring its considerable scientific and application potential. While the experimental design is rigorous and well-supported by data, further refinement is needed in the theoretical mechanism analysis and comparative discussion with existing literature.

We thank the Reviewer for recognizing that our study “achieves a significant breakthrough in maintaining antiferroelectricity even at a thickness of 9 nm”.

1. The manuscript lacks an in-depth analysis of the potential impact of the LaLuO₃ buffer layer on interfacial defects and the long-term stability of lattice mismatch. It is recommended to supplement the study with high-resolution STEM analysis of the interface structure or theoretical calculations, such as interfacial energy and stress distribution.

Following the recommendation from the Reviewer, we now provide additional information on the structure of the films with high-resolution STEM analysis of the interface structure (Supplementary Figure 2) and complementary inputs on the strain profiles (Supplementary Figure 5 and Supplementary Figure 6). Please see our replies to Points 1-2 of Reviewer 2 and Points 2-4 of Reviewer 3 for details.

We also agree that a theoretical study of the interface could shed light on the impact of the LaLuO₃ (or, rather, the LaLuO₃/SrPbO₃) buffer layer. However, we must stress that this would be a highly nontrivial investigation that lies beyond the scope of the present work, for several reasons:

First, we do not have access to information about the interatomic couplings in the PbZrO₃/SrPbO₃ interface – be they experimental or theoretical –, which means that any theoretical study would have to rely on a predictive quantum mechanical approach (e.g., Density Functional Theory or DFT), which imposes stringent limitations with regard to the size of the simulated systems. More precisely, for a material like PbZrO₃, where the relevant energy differences separating competing polymorphs are around 1 meV/f.u. [<https://doi.org/10.1038/s41524-021-00671-w>] meaning that extremely accurate DFT calculations are thus required, the simulations would be restricted to periodic supercells of 100 or 200 atoms at most.

Second, a theoretical investigation of interfacial defects – for such a novel interface and in the absence of experimental guidelines – is a daunting task. On the one hand, it would require simulation supercells exceeding what is possible at the DFT level. On the other hand, the number of possibilities to consider is enormous even if we restrict ourselves to point defects. (Which kind of vacancy or vacancy + substitutional combination? Antisite defects or interdiffusion? Which charge state of the defect?) Consequently, performing a DFT simulation that is relevant to the experimental system is, at best, highly uncertain.

Finally, we point out that, to the best of our knowledge, theoretical investigations of the kind suggested by the Referee are all but absent from the literature — particularly for a compound as complex as PbZrO₃ and an interface as novel as PbZrO₃/SrPbO₃. For all these reasons, while admittedly very interesting, such an investigation is out of the scope of the present work.

2. The experiment revealed a significant increase in the critical electric field as the film thickness decreased (from 0.3 MV/cm to 2.4 MV/cm). However, the study merely hypothesizes that “the

nucleation of the ferroelectric phase is hindered in the pure antiferroelectric matrix." It is recommended to incorporate phase-field simulations or first-principles calculations to quantitatively analyze the effect of thickness on the phase transition energy barrier.

Here too, we agree with the Referee that performing such phase-field or first-principles simulations would be very valuable. However, running relevant simulations is currently not feasible, for several reasons. Let us note that, for such a theoretical study to be relevant, one should be able to:

- (1) have a realistic representation of the SrPbO₃/PbZrO₃ interface, including the defects that probably act as nucleation centers for a field-driven transformation;
- (2) consider simulation supercells large enough to allow for nucleation and growth of the field-favored phase;
- (3) consider simulation supercells large enough to allow for a discussion of the thickness dependence of the transformation;
- (4) be able to simulate the material under the action of an external electric field.

In the context of DFT simulations (and as mentioned in our reply to Point 1 above), all four points become problematic, as we are severely restricted as regards the size of the simulation supercell, we have no information about the interfacial defects that are likely to have a large impact on how the transformation proceeds, and it is by no means trivial to consider the action of an electric field in a metal-insulator-metal system.

Thus, this is an extremely challenging problem from the DFT perspective, as consistent with the fact that there are no precedents in the literature, as far as we can tell. If by contrast we consider a phase-field simulation, the main challenge is to construct a Ginzburg-Landau model of the Pt/PbZrO₃/SrPbO₃ system. Let us stress that constructing a good (predictive) continuum model of PbZrO₃ is a daunting task in itself, because of the large number of relevant order parameters and the complexity of their cross-couplings. While there has been some progress (see, e.g., <https://doi.org/10.1103/PhysRevMaterials.7.L071401>), such models are not available to us and not compatible with the usual phase-field implementations. If we add the difficulty of extending the Ginzburg-Landau model to the SrPbO₃ electrode and the interface (as essential to understand the presumed importance of the interfacial couplings), the task becomes even more challenging. For all these reasons, while admittedly very interesting, such an investigation is out of the scope of the present work.

3. Although the study cites theoretical predictions (e.g., Reference 20) supporting the stabilization of the antiferroelectric phase under tensile strain, it does not incorporate computational data (e.g., strain-phase diagrams) to quantitatively validate the experimental results.

4. Compared to recent similar studies (e.g., References 17 and 18, which report the coexistence of ferroelectric phases under compressive strain), a more detailed discussion is needed on the physical mechanisms through which strain direction influences the stabilization of ferroelectric and antiferroelectric phases in thin films.

Let us address these two comments together. Following this comment from the Referee, we have performed simulations – employing an accurate machine-learned force field (MLFF) derived from DFT data – addressing the question of the stability of the antiferroelectric phase as a function of epitaxial strain. Following the standard approach in DFT studies of thin films, we consider pure PbZrO₃ as we would in a simulation of the bulk compound – i.e., periodically repeated in all spatial directions – while imposing the epitaxial constraint by fixing the in-plane lattice vectors and letting all atoms and out-of-plane lattice vector relax. Our results are summarized in Fig. R1, which shows the energy of the antiferroelectric polymorph (for three different orientations with respect to the substrate plane) with respect to the ferroelectric phase (which is also resolved as a function of the substrate's lattice constant). The curves in the figure correspond to the results obtained when a square substrate is considered. Finally, the isolated symbols correspond to the results obtained when the epitaxial

condition corresponding to a square substrate is modified to account for the in-plane anisotropy imposed by LaLuO_3 . Several interesting conclusions can be derived from these results.

First, we see that the antiferroelectric state is generally favored for substrate lattice constants between 4.05 and 4.28 Å, although the ferroelectric phase remains in competition. Second, there is a clear difference depending on the orientation of the antiferroelectric structure with respect to the plane of the substrate. The ac and bc orientations (the experimentally observed ones, which are equivalent by symmetry for an isotropic epitaxial constraint) are favored for square substrates with $a_0 \sim 4.06$ Å, while the ab orientation dominates for substrates with $a_0 \sim 4.20$ Å. Note that the ab orientation corresponds to having the antipolar distortions in the plane of the substrate, a feature that – according to our simulations – seems to be favorable as we move toward greater tensile strains and which is compatible with the general notion that, in ferroelectric perovskites, the polar distortions always align with the long lattice constant. This same reasoning is consistent with the fact that the ferroelectric phase – featuring a homogeneous polar distortion – becomes dominant in both limits, i.e., for very large and very small a_0 values.

Fig. R1. a, Three Pbam antiferroelectric polymorphs of PbZrO_3 (ac, ac, bc) are compared with the R3c ferroelectric phase as a function of in-plane strain. b, Energy difference between the antiferroelectric (AFE) polymorph with ac, bc, ab orientations and the ferroelectric (FE) phase R3c, computed as a function of the pseudo-cubic in-plane lattice, considering biaxial strain. The isolated symbols show calculation results considering the non-equal in-plane parameters of $\text{LaLuO}_3(110)_o$. Note that including the anisotropic lattice parameters makes the antiferroelectric polymorphs considerably more energetically favoured relative to isotropic lattice parameters, as reflected by the data points markedly below the respective lines. c, Angle of the local dipoles with respect to the in-plane direction in the ferroelectric phase and the antiferroelectric phase for ab (AFE ip) and ac, bc (AFE oop) orientations.

For completeness and to better illustrate this point, in Fig. R1c we show the evolution of the angle that the local dipoles – associated to the Pb atoms – form with the plane of the square substrate. The results for the ferroelectric phase evidence a rotation of the polarization, which lies in the plane of the substrate for sufficiently large tensile strain ($a_0 > 4.23$ Å). The results for the antiferroelectric phase depend, naturally, on the orientation: the angle is always zero for the “AFE ip” case (equivalent to ab,

where the dipoles lie within the substrate) while they rotate only slightly towards the plane for the “AFE oop” case (ac and bc). The latter result is interesting, as it suggests that the complex structure of the antiferroelectric phase – which features a non-trivial mixture of oxygen octahedra tilts and antipolar distortions – confers some rigidity to the dipole structure. These subtle and intricate aspects remain a task for future investigation.

Thus, interestingly, our MLFF simulations predict that epitaxial films grown on LaLuO₃ should present an antiferroelectric structure, but with an orientation that differs from the one we experimentally observe in our samples. These results suggest that the structural couplings across the SrPbO₃/PbZrO₃ interface – or other structural features absent in our idealized (defect free) PbZrO₃ simulations – must be responsible for stabilizing the antiferroelectric phase in the orientation observed in our samples.

5. The image annotations in the manuscript should be clearer (e.g., STEM labeling in Figure 1). Additionally, the Methods section should provide a more detailed description of certain parameters, such as the influence of laser energy density on film crystallinity, to enhance experimental reproducibility.

We have improved our labelling in Figure 1. Regarding PbZrO₃ growth, lower energy density and/or higher temperature give rise to non-stoichiometric precipitates on the film surface. Following the recommendation of the Reviewer, we added information on the influence of temperature and laser energy density in the Methods section:

“The sample temperature was then reduced to 560°C, while O₂ was replaced by N₂O with dynamic pressures of 40 mTorr and 120 mTorr for SrPbO₃ and PbZrO₃, respectively. *For both materials, higher growth temperatures result into high lead-deficiency and increased surface roughness.*”

“(…) and these values were then increased to 5 Hz and 2.5 J/cm² for the PbZrO₃ layer. *Lower fluence gives rise to non-stoichiometric precipitates on the film surface.*”

6. According to the reference (Phys. Rev. Lett. 2015, 115, 097601), the critical thickness for AFE PZO thin films is about 6.5 nm. Whether a thickness below 9 nm can remain stay a AFE structure?

In the reference mentioned by the Reviewer, the critical thickness of 6.5 nm is calculated for short circuit boundary conditions and zero strain. The same authors show that under tensile strain (2%), films with 5 nm thickness remain antiferroelectrics (Suppl. Information of the same Phys. Rev. Lett. Paper). Instead, under ideal open circuit conditions, the large depolarizing field prohibits the stabilization of the ferroelectric phase and maintains the antiferroelectric phase without any thickness limitations.

7. In Figure 1, I cannot find the thickness information about this picture. The author should describes it in the STEM images. A 9-nm-thickness HAADF-STEM image may be more valuable than other thicknesses.

As written in the main text of the manuscript, Figure 1 shows a cross-section STEM image of the epitaxial stack “... with thicknesses of 30 nm, 19 nm, and 108 nm for PbZrO₃, SrPbO₃, and LaLuO₃, respectively.”

We added this information into the caption of Figure 1 to make it clearer to the reader:

“*The layer thicknesses are 30 nm, 19 nm, and 108 nm for PbZrO₃, SrPbO₃, and LaLuO₃, respectively.*”

8. In Figure 2a, XRD data only show one thickness, but the thickness was not given, and other thicknesses were lacking. Moreover, the 2θ XRD pattern that starts from 38 to 50 degree is enough.

We revised the caption of Figure 2 to add the thickness information:

“a, 2θ - ω X-ray diffraction pattern showing the $(hh0)_o$ symmetric peaks of the DyScO_3 , LaLuO_3 and SrPbO_3 as well as the $(h,2h,0)_o$ peak for PbZrO_3 , with $h = 1, 2, 3, 4$ for layer thicknesses of 30 nm, 19 nm, and 108 nm for PbZrO_3 , SrPbO_3 , and LaLuO_3 , respectively. The small peak around 40° corresponds to the Pt(111) of the capacitors. b, Reciprocal space map around $\text{DyScO}_3(332)_o$ for the same sample as in (a). The three epitaxial layers of LaLuO_3 , SrPbO_3 , PbZrO_3 share a common in-plane lattice parameter.”

Regarding X-ray characterization for other PbZrO_3 film thicknesses, we have provided the reciprocal space maps in Supplementary Figure 10 and Supplementary Figure 11 for four different films thicknesses (9, 20, 40, 200 nm). In addition, we now show the 2θ - ω X-ray diffraction pattern between 40 and 49 degrees for all the film thicknesses, i.e., 9, 20, 30, 40, 80, 200 nm (Fig. R2). This Figure has been added as Supplementary Figure 7 and we have modified the sentence page 5 accordingly:

“A typical 2θ - ω X-ray diffraction pattern is displayed in Figure 2a, showing high crystalline quality for the LaLuO_3 , SrPbO_3 and PbZrO_3 thin films and no secondary phases (other film thicknesses are displayed in Supplementary Figure 7).”

Fig. R2. (Supplementary Fig. 7 of the revised manuscript) 2θ - ω X-ray diffraction patterns of the $\text{PbZrO}_3/\text{SrPbO}_3/\text{LaLuO}_3$ samples grown on DyScO_3 , with PbZrO_3 film thicknesses varying from 200 nm (bottom) to 9 nm (top).

9. In Figure 3b, the P-E loops of 9-nm-thickness is lacking, even through the author explain that curves cannot be obtained due to a capacitive contributions.

The four switching peaks associated with antipolar/transitions in the I-V loops (Fig. 3a) and the remanent antipolar ordering observed by STEM (Fig. 4c) clearly support our interpretation of a preserved antiferroelectricity down to 9-nm thickness. We only want to show raw data for the I-V and P-V loops recorded at 2 kHz on these Pt/PbZrO₃/SrPbO₃ capacitors, in order to avoid possible controversies (e.g., “Ferroelectrics go bananas, J.F. Scott, J. Phys. Cond. Matt. 20, 021001 (2008)). Integrating the I-V of Figure 3 for the 9-nm-thick PbZrO₃ layer, after removal of the parasitic contribution, would only provide unnecessary and inaccurate information.

Nevertheless, we manually removed this parasitic background and integrated the remaining peaks (Fig. R3).

Fig. R3. Parasitic background removal in the J-V curve of the Pt/PbZrO₃/SrPbO₃ capacitors for the PbZrO₃ thickness of 9 nm. a, Raw J-V in black and parasitic contribution in red. b, Integrated P-V loop from the difference between the black and red curves.

10. In page 3 line 98, there are some unforetold description that (we expect PbZrO₃ to grow under an anisotropic epitaxial tensile strain of +0.4% and +1.2-1.8% in the two in-plane directions), did the special tensile value is suitable for the stability of AFE phase?

Considering (120)_o PbZrO₃ on top of (110)_o SrPbO₃/LaLuO₃ with all the three orthorhombic layers having their c_o parallel, we can estimate the strain imposed by the orthorhombic layers on PbZrO₃. Taking the bulk cell parameters of PbZrO₃, LaLuO₃, and SrPbO₃, the strain is estimated along c_o or perpendicularly to c_o as:

	PbZrO ₃	LaLuO ₃	SrPbO ₃
ao (Å)	5.882	5.810	5.852
bo (Å)	11.783	6.013	5.969
co (Å)	8.228	8.373	8.324
// c _o (Å)	4.114	4.187	4.162
⊥ c _o (Å)	4.163	4.181	4.180
strain PbZrO ₃ // c _o (%)		1.76	1.17
strain PbZrO ₃ ⊥ c _o (%)		0.44	0.41

Table R1. Bulk orthorhombic unit-cells of PbZrO₃, SrPbO₃, and LaLuO₃ and estimated strain on PbZrO₃.

Here the strain is simply calculated as $(x\text{PZO}-x\text{LLO(SPO)})/(x\text{PZO})$. Hence, we clearly see that the in-plane strain imposed by LaLuO_3 or SrPbO_3 on PbZrO_3 is anisotropic, with +0.4% and +1.2-1.8%, perpendicularly or along c_0 , respectively. This is due to the fact that the lattice mismatch between PbZrO_3 and SrPbO_3 or LaLuO_3 is larger along c_0 , than along a_0 or b_0 . To the best of our knowledge, there is no published input on the impact of such anisotropic tensile strain on the stability of the antiferroelectric phase. From the simulations presented in Fig. R1, we can qualitatively argue that deviating from a purely cubic substrate seems to enhance the stability of the antiferroelectric phase.

11. In page 4 line 164, (Such large critical fields are consistent with measurements in pure PbZrO_3 single crystals) should provide a new reference to support the statement.

We added a new reference (Fesenko, O. E., Kolesova, R. V. & Sindeyev, Yu. G. The structural phase transitions in lead zirconate in super-high electric fields. *Ferroelectrics* **20**, 177–178 (1978)) to support our statement that the "large critical fields are consistent with measurements in pure PbZrO_3 single crystals". In this reference, measurements of P-E loops in single crystals of PbZrO_3 indicate critical fields of the order of 300-400 kV/cm.

12. The supplementary was not mentioned in the manuscript, the author may explain it at proper position.

All the Figures from the Supplementary Information are referenced in the text as "Supplementary Figure 1-12". We added a note to reference the Supplementary Information after the Methods.

13. In the bottom inset of Figure 4a, the altitude of the red antipolar pattern is quite higher than the blue one. Does these structures are uncompensated or the fitting algorithm is accurate enough? The author can explain this.

Several experimental constraints can limit the accuracy of polar displacement determination, such as crystal misalignment, residual aberrations, limited signal-to-noise ratio, and fitting accuracy. Nevertheless, we believe that at the nanoscale, polarization compensation may be locally incomplete. For instance, such an evolution is visible in Figure 4 when approaching the surfaces. Furthermore, previous studies have also reported partially uncompensated polarization in perovskite antiferroelectrics (<https://doi.org/10.1103/PhysRevLett.123.217602>).

14. In Figure 5, why the antipolar pattern is not stable in the 9-nm-thick PbZrO_3 film or it is general phenomena? A theoretical word to explain the dynamics is also lacking.

As we wrote in the manuscript, the antipolar pattern of thicker films (20 or 30 nm) was robust against electron beam exposure. Further investigations should be realized to determine the precise role of the interfaces on the nucleation of translational boundaries in such ultrathin epitaxial films.

Reviewer #2 (Remarks to the Author):

This manuscript presents an elegant and well-executed study on the stabilization of a pure antiferroelectric phase in ultrathin PbZrO_3 films under tensile epitaxial strain. The authors convincingly demonstrate that careful strain engineering using LaLuO_3 -buffered DyScO_3 substrates allows the realization of an antipolar ground state even in films as thin as 9 nm. The combination of high-quality thin film synthesis, detailed structural characterization using scanning transmission electron microscopy (STEM), and electrical measurements makes a compelling case for the emergence of antipolar instabilities as a function of thickness. The work is timely and addresses a central challenge in the field—namely, the stabilization of nonpolar or antipolar states at reduced dimensions, which is

of great interest for antiferroelectric-based applications. Moreover, the observation of distinct dipolar patterns, and their evolution under electron beam irradiation, opens up new questions regarding the interplay between external perturbations and intrinsic lattice instabilities. Overall, I believe this study makes a meaningful contribution to the ongoing effort to understand and control ferroic orders in reduced dimensions and complex oxide heterostructures. However, I suggest the authors address the following points to further improve the clarity, completeness, and rigor of the manuscript:

We thank the Reviewer for their positive comments. In the following, we address the questions they raised point by point.

1. Clarification of strain relaxation in the buffer layer stack: The authors use SrPbO₃/LaLuO₃-buffered DyScO₃ substrates to impose tensile strain in PbZrO₃ films. Given the crucial role of strain in stabilizing the antiferroelectric phase, it would be helpful if the authors could elaborate on how strain evolves across the DyScO₃ substrate, LaLuO₃, and SrPbO₃ layers. In particular, considering that both LaLuO₃ and SrPbO₃ serve as tensile buffer layers, and that SrPbO₃ is known to be vulnerable to electron beam exposure, further clarification is needed on the specific function and necessity of including SrPbO₃, rather simply using LaLuO₃, in the buffer stack.

We first considered the growth of PbZrO₃ directly on LaLuO₃. As illustrated by the HAADF-STEM images in Fig. R4, PbZrO₃ grows perfectly on the LaLuO₃ orthorhombic structure and shows no evidence of strain relaxation at the interface. However, to be able to perform electrical measurements, we had to interface the PbZrO₃ with an epitaxial oxide bottom electrode. We chose SrPbO₃ since it is well lattice matched to LaLuO₃ and PbZrO₃.

Fig. R4. STEM investigations of PbZrO₃/LaLuO₃ bilayers grown on DyScO₃(110)_o. a, HAADF-STEM and b, BF-STEM images. Horizontal scale bars are 5 nm.

The relaxation of the LaLuO₃ film occurs primarily at the LaLuO₃/DyScO₃ interface. This is now more clearly visible thanks to the addition of the out-of-plane and in-plane strain profiles shown in Supplementary Figure 5-6 (Fig. R5, R6). As demonstrated in these new Supplementary Figures, the relaxation occurs in a similar manner regardless of whether the SrPbO₃ electrode remains crystalline or becomes strongly amorphized by the electron beam.

Fig. R5. (Supplementary Fig. 5 of the revised manuscript) a, Bright field STEM image (left) on the cross-section specimen of the PbZrO_3 (20 nm) / SrPbO_3 (15 nm) / LaLuO_3 (110 nm) epitaxial stack grown on $\text{DyScO}_3(110)_o$. Geometrical phase analysis gives rise to the b, out-of-plane (middle) and c, in-plane (right) deformations at the unit-cell level. The corresponding out-of-plane and in-plane deformation profiles are displayed in (d) and (e), respectively. The constant in-plane deformation between the three layers suggests that PbZrO_3 is coherently strained by the LaLuO_3 film.

Fig. R6. (Supplementary Fig. 6 of the revised manuscript) a, Bright field STEM image (left) on another cross-section specimen of the PbZrO_3 (20 nm) / SrPbO_3 (15 nm) / LaLuO_3 (110 nm) epitaxial stack grown on $\text{DyScO}_3(110)_o$. In this area, the SrPbO_3 electrode has been strongly amorphized by the electron beam. Geometrical phase analysis gives rise to the b, out-of-plane (middle) and c, in-plane (right) deformations at the unit-cell level. The corresponding out-of-plane and in-plane deformation profiles are displayed in (d) and (e), respectively. The constant in-plane deformation between the layers suggests that PbZrO_3 is still coherently strained by the LaLuO_3 film.

We have modified the manuscript page 5 to include this discussion:

“From the geometrical phase analysis (Fig. 1c-d, Supplementary Figure 4), we observe that the large lattice mismatch strain (> 5%) between LaLuO_3 and DyScO_3 is relieved within the first 10 nm of the layer.

Additional strain profile analyses taken on different areas with weakly (Supplementary Figure 5) or strongly amorphized SrPbO_3 layers (Supplementary Figure 6) show that LaLuO_3 , SrPbO_3 , and PbZrO_3 display a constant in-plane lattice parameter value, suggesting that the epitaxial stack is coherently strained to the LaLuO_3 buffer layer.”

2. STEM characterization of the $\text{PbZrO}_3/\text{SrPbO}_3$ interface: Despite the e-beam sensitivity of SrPbO_3 , high-resolution STEM images of the $\text{PbZrO}_3/\text{SrPbO}_3$ interface—either immediately post-FIB or after TEM sample preparation—would be highly informative. Such data would help clarify the microstructural integrity and strain transfer across the interface, and allow readers to better assess the role and effectiveness of the tensile strain buffer layers.

We thank the Reviewer for sharing the information that “ SrPbO_3 is known to be vulnerable to electron beam exposure” as we were not aware that the SrPbO_3 electron beam sensitivity was a general trend. The Reviewer is perfectly right in requiring local investigations of the $\text{PbZrO}_3/\text{SrPbO}_3$ interface, and we have been able to obtain this information. In the Figure R7, we provide local STEM investigations of the SrPbO_3 layer in the few areas we could find where it remains crystallized in the cross-section specimen. We first confirm the X-ray diffraction results and demonstrate that SrPbO_3 grows with $(110)_o$ orientation and the c_o axis parallel to that of DyScO_3 , as does the LaLuO_3 orthorhombic perovskite (Fig. R7a-b). Furthermore, we clearly observe the continuity of the planes at the $\text{SrPbO}_3/\text{LaLuO}_3$ and $\text{PbZrO}_3/\text{SrPbO}_3$ interfaces, as emphasized by the HAADF-STEM image in Fig. R7c. In the revised version of the manuscript, we added this Figure as Supplementary Figure 2 and referenced it to the manuscript in the sentence, page 5:

“Nevertheless, local observations confirm that SrPbO_3 grows with the same $(110)_o$ orientation as the LaLuO_3 and DyScO_3 (Supplementary Figure 2).”

Fig. R7. (Supplementary Fig. 2 of the revised manuscript) STEM investigations of the SrPbO_3 layer. a, ABF-STEM image on a cross-section area with the zone axis parallel to the c_o axis of DyScO_3 . FFTs from the LaLuO_3 and SrPbO_3 layers indicate that they both have an orthorhombic structure with their c_o axes parallel to that of the DyScO_3 substrate. b, BF-STEM image on a cross-section area with the zone axis

perpendicular to the c_o axis of DyScO_3 . FFTs for LaLuO_3 and SrPbO_3 indicate that their c_o axes lie in the film plane. c, HAADF-STEM image on a cross-section area showing the $\text{LaLuO}_3/\text{SrPbO}_3/\text{PbZrO}_3$ interfaces. Despite some partial amorphization at the $\text{LaLuO}_3/\text{SrPbO}_3$ interface, the continuity between the planes is clearly seen between the three layers with a sharp Z-contrast inversion between the SrPbO_3 and PbZrO_3 , as the heavy Pb element jumps between the B-site and the A-site. Horizontal scale bars are 5 nm.

3. Visualization of polar configurations in HAADF-STEM: In Figures 4 and 5, the up-up-down-down and other polar configurations are illustrated using numerous blue and red arrows. While this visual aid is helpful in identifying local dipole arrangements, the use of extensive artificial annotations without a clear explanation of how the dipole directions are extracted—may be misleading for readers unfamiliar with the technique. The authors are encouraged to explicitly describe the methodology used for determining dipole orientations (e.g., Atomap, Gaussian fitting, or other image processing tools), and to provide side-by-side versions of the images with and without the arrow overlays for better comparison and interpretation.

To determine the dipole orientations, we have fitted the intensity maxima positions in the HAADF-STEM images using two-dimensional Gaussian functions, as available within the “Atomap” processing tools. The off-center displacements of Pb cations were calculated with respect to the geometric center of the four surrounding Zr-site columns and the arrows representing the off-center intensities and directions were overlaid on the HAADF images at the Pb column positions.

We modified the Methods accordingly: *“Polar displacements were evaluated using two-dimensional Gaussian fitting of the atomic positions in the HAADF images, as implemented in the Atomap software³⁵. The off-center displacements of Pb cations were calculated with respect to the geometric center of the four surrounding Zr-site columns, and the arrows representing the off-center intensities and directions were overlaid on the HAADF images at the Pb column positions.”*

This technique has been previously reported, for instance, in Phys. Rev. Lett. **123**, 217602 (2019) « Uncompensated Polarization in Incommensurate Modulations of Perovskite Antiferroelectrics” (<https://doi.org/10.1103/PhysRevLett.123.217602>) or *Appl. Phys. Rev.* **10**, 021405 (2023) “Ferroelectric phase transitions in epitaxial antiferroelectric PbZrO_3 thin films” (<https://doi.org/10.1063/5.0143892>). We have also cross-checked the establishment of the antipolar texture in PbZrO_3 thin films using annular bright-field imaging contrast in scanning transmission electron microscopy (ABF-STEM). Although ABF imaging is more prone to artifacts associated with residual aberrations, it is sensitive to the positions of light elements such as oxygen. Figure R8 below shows ABF data obtained for a 9 nm PbZrO_3 film. Fig. R8a confirms the cation displacements in agreement with the HAADF-STEM images. Fig. R8b shows that the positions of the oxygen atoms in the Pb rows are consistent with the antiferroelectric (AFE) configuration (i.e., located at the left-lower, center, and right-upper positions with respect to the Pb–Pb pairs). A representative ABF-STEM image is shown in Fig. R8c and compared with the projected PbZrO_3 structure.

Fig. R8. a, ABF-STEM image with inverted contrast. The polar displacements were determined by comparing the position of Pb atoms relative to the barycenter of the Zr square lattice. The resulting dipoles are represented by red and blue arrows as a function of their directions along the vertical axis. b, ABF-STEM image with inverted contrast. The displacement of oxygen atoms relative to the barycenter of the two neighboring Pb atoms along the horizontal line is indicated by arrows. Left and right displacements are encoded in blue and red, respectively. When the measured lateral displacement is below 14 pm, the arrow is displayed in yellow. c, Zoomed-in ABF-STEM image with superimposed atomic models.

Following the recommendations from the Reviewer, we now provide in the Supplementary Information (Supplementary Figure 12) and in Fig. R9 the raw HAADF-STEM images side-by-side with overlapped vector maps obtained in Figure 4.

Fig. R9. (Supplementary Fig. 12 of the revised manuscript) High resolution HAADF-STEM images for a, the 30-nm-thick PbZrO_3 film, b, the 20-nm-thick PbZrO_3 film and c, the 9-nm-thick PbZrO_3 . In all the images, the zone axis is parallel to the c_0 axis of DyScO_3 . The resulting dipoles are represented as coloured arrows in the right images (same images as in Figure 4). Horizontal scale bars are 5 nm.

4. Reversibility of electron beam–induced changes: The electron beam–induced transformation of the antipolar texture in the 9-nm-thick PbZrO_3 film is fascinating. However, the reversibility of this phenomenon remains unclear. Can the authors clarify whether the antipolar pattern reappears after the electron beam is turned off for a sufficient period? This would help determine whether the beam-induced changes are reversible or represent a permanent structural transition. A brief discussion of this point in the main text would be helpful.

We do agree with the Reviewer’s remark on the “fascinating” and unexpected electron beam impact on the antiferroelectric ordering at low thickness. This would deserve a dedicated study with pump-probe experiments, and/or with a wedged thin film to get relevant insights on such antiferroelectric instabilities, beyond the scope of the manuscript.

In summary, this is a carefully conducted and high-quality study that advances our understanding of antiferroelectricity in strained oxide films. I recommend minor revisions to address the points above, which would further strengthen the manuscript’s clarity and impact.

We thank the Reviewer for their comments and clarification requests. We believe the manuscript is now improved following their inputs.

Reviewer #3 (Remarks to the Author):

The polar structures and final responses of PbZrO₃ (PZO) under different external fields are classic issues in the antiferroelectric field. Since the lattice parameters of PZO are significantly larger than those of commonly used perovskite oxide single crystal substrates, it is difficult to obtain coherently strained PZO films. In this manuscript, the authors have grown LaLuO₃ (LLO) layer on the DyScO₃ (DSO) substrate, which is almost fully relaxed thus the large lattice of LLO was preserved. And they claim that the PZO films can be coherently strained on this buffer layer, thus a “pure” antiferroelectric PZO phase can be obtained.

While the work is successful in obtaining LLO as a buffer layer with large lattice parameters, the results regarding the PZO films are not sufficiently remarkable. It is worth noting that the approach of using a buffer layer to obtain high - quality PZO films or the pure-phase PZO has been reported in previous studies, as cited in References 4 and 23. Moreover, these earlier works not only achieved the ideal antiferroelectric PZO films but also utilized PZO in high - performance thermal switching chips or electromechanical devices. Therefore, the novelty of the current work is insufficient for publication in Nature communications.

The results from Liu et al. published in Science (Ref. 23) are very interesting, but they are based on relaxed 150-nm-thick PbZrO₃ thin films grown on highly mismatched SrRuO₃-buffered SrTiO₃ substrates. Therefore, we disagree with the Reviewer on the similarity with our manuscript, as our approach deals with the growth of PbZrO₃ on SrPbO₃-buffered LaLuO₃, both materials being well lattice matched with PbZrO₃. The results from Pan et al. published in Nature Materials in 2024 (Ref. 4) are also outstanding and also deal with thick PbZrO₃ films (100 nm). They show that the clamping of the antiferroelectric material thanks to epitaxy gives rise to a very large electromechanical response. Hence, none of these two papers focus on the experimental influence of epitaxial tensile strain on the antiferroelectric properties of PbZrO₃ down to the ultrathin limit. Here, the key novelty in our work is our focus on PbZrO₃ film thicknesses of 9, 20, 30 nm under tensile epitaxial strain, and how we show that this strain gives rise to properties that are distinct from those already reported in the literature, both regarding the antipolar pattern characteristics and the P-E double hysteresis loops.

Technically, there are some obvious details seem indicating that the phases here in PZO are not “pure”:
1. In Figure 1b, distinct strip domains are visible within the PZO layer, yet the authors have not provided any discussion on these features. The presence of these domains implies that the PZO films are not in a pure state.

Figure 1b indeed shows some stripe-like features in the PbZrO₃ layer. These features correspond to translation boundaries, and the presence of similar boundaries has already been reported and discussed in antiferroelectric (AFE) materials (e.g., Wei et al., *Materials Research Bulletin* 62, 2015, p. 101 (<https://doi.org/10.1016/j.materresbull.2014.11.024>)). On the other hand, stripes have been also observed in the case of coexisting ferrielectric and antiferroelectric phases (Yu, Z. et al., *Room-temperature stabilizing strongly competing ferrielectric and antiferroelectric phases in PbZrO₃ by strain-mediated phase separation*, *Nat. Commun.* **15**, 3438 (2024), Ref. 12). Nevertheless, in our case PbZrO₃ remains within a pure *Pbam* phase, exhibiting a $\frac{1}{4}$ super periodicity characteristic of the antipolar ordering. The spatial influence of the stripes is very limited as for instance, no other periodicities—such as a $\frac{1}{2}$ periodicity corresponding to ferrielectric order—have been observed. Fig. R10 shows the same area as shown in Figure 1b, along with a Fast Fourier Transform (FFT) taken from an approximately 150 × 150 nm² region within a single antipolar domain. The FFT spots are sharp (Fig. R10b), and the intensity profile across the super periodicity peaks clearly shows only the $\frac{1}{4}$ periodicity (Fig. R10c), with no indication of other – even faint – modulations.

Fig. R10. (Supplementary Fig. 3 of the revised manuscript) a, Same image as in the main text Figure 1b with layer thicknesses of 30 nm, 19 nm, and 108 nm for PbZrO_3 , SrPbO_3 , and LaLuO_3 , respectively. b, FFT from the squared area in (a), located in the green PbZrO_3 domain. c, Intensity profile from the FFT pattern, showing a well-defined $\frac{1}{4}$ super periodicity corresponding to a pure $Pbam$ phase.

The following example considers an area where more translation boundaries are present (Fig. R11). The FFT pattern reveals a super periodicity with more elongated features (Fig. R11b), consistent with the increased structural disorder. We performed geometric phase analysis (GPA) on this super periodicity, and the resulting phase image is displayed in Fig. R11c. The phase intensity profile across the boundaries indicates an almost pure π -phase shift occurring within a unit-cell scale (Fig. R11d). Once again, this demonstrates that the presence of such translational boundary line defects does not contradict the existence of a pure $Pbam$ antiferroelectric phase in-between them.

Fig. R11. (Supplementary Fig. 3 of the revised manuscript) a, HAADF-STEM image of an area containing several “stripe lines” in a 30 nm PbZrO_3 layer. b, Corresponding FFT pattern; the red circle indicates the $\frac{1}{4}$ periodicity selected for phase analysis. c, Phase distribution of the super periodicity; the blue and yellow regions are separated by a π -phase shift. d, Phase profile across the antiphase domain and translation boundaries (in radians), taken from the boxed area in (c).

We added the following sentence to the manuscript, page 5: “The presence of dark lines in the PbZrO_3 domains (Fig. 1b) is attributed to antiphase boundaries, but the film remains purely in the antipolar $Pbam$ phase (Supplementary Figure 3).”

2. Since the LLO layer is thicker, and is fully relaxed, there should be lots of threading dislocations inside the LLO layer. Moreover, these threading dislocations should be reproduced no matter what layer was grown on it (when coherent along the in-plane direction). Where are these threading dislocations? Why they seem doesn’t affect the responses of PZO? Note that the mismatch here for LLO and DSO is large, which should produce lots of threading dislocations connecting with the misfit dislocations (as illustrated in Fig. S1).

The relaxation of the LaLuO_3 film occurs primarily at the $\text{LaLuO}_3/\text{DyScO}_3$ interface. This is now more

clearly visible thanks to the addition of the out-of-plane and in-plane strain profiles shown in Fig. R12, R13 and Supplementary Figure 5-6.

Fig. R12. (Supplementary Fig. 5 of the revised manuscript) a, Bright field STEM image (left) on the cross-section specimen of the PbZrO_3 (20 nm) / SrPbO_3 (15 nm) / LaLuO_3 (110 nm) epitaxial stack grown on $\text{DyScO}_3(110)_o$. Geometrical phase analysis gives rise to the b, out-of-plane (middle) and c, in-plane (right) deformations at the unit-cell level. The corresponding out-of-plane and in-plane deformation profiles are displayed in (d) and (e), respectively. The constant in-plane deformation between the three layers suggests that PbZrO_3 is coherently strained by the LaLuO_3 film.

Fig. R13. (Supplementary Fig. 6 of the revised manuscript) a, Bright field STEM image (left) on another cross-section specimen of the PbZrO_3 (20 nm) / SrPbO_3 (15 nm) / LaLuO_3 (110 nm) epitaxial stack grown on $\text{DyScO}_3(110)_o$. In this area, the SrPbO_3 electrode has been strongly amorphized by the electron beam. Geometrical phase analysis gives rise to the b, out-of-plane (middle) and c, in-plane (right) deformations at the unit-cell level. The corresponding out-of-plane and in-plane deformation profiles are displayed in (d) and (e), respectively. The constant in-plane deformation between the layers suggests that PbZrO_3 is still coherently strained by the LaLuO_3 film.

As demonstrated in these new Supplementary Figures, the relaxation occurs in a similar manner regardless of whether the SrPbO₃ electrode remains crystalline or becomes strongly amorphized in the cross-section specimen. To gain more detailed insight into the interfacial relaxation, geometrical phase analysis (GPA) was performed at higher magnification near the LaLuO₃/DyScO₃ interface. As shown in Fig. R14, most dislocations appear at the interface, indicating a semi-coherent relationship between the LaLuO₃ film and the DyScO₃ substrate. A few dislocations are also visible approximately 10 nm away from the interface, contributing to further strain relaxation. Further away from the interface, the LaLuO₃ can be considered fully relaxed, with relatively homogeneous in-plane and out-of-plane lattice parameters.

Fig. R14. Strain relaxation of the LaLuO₃ thin film on the DyScO₃ substrate. (left) HAADF-STEM image of the LaLuO₃/DyScO₃ area. GPA analysis of the (middle) out-of-plane and (right) in-plane deformations. The dashed blue line (left) indicates the localization of the LaLuO₃/DyScO₃ interface.

We have modified the manuscript, page 5, to take these comments into account:

“From the geometrical phase analysis (Fig. 1c-d, Supplementary Figure 4), we observe that the large lattice mismatch strain (> 5%) between LaLuO₃ and DyScO₃ is relieved within the first 10 nm of the layer. Additional strain profile analyses taken on different areas with weakly (Supplementary Figure 5) or strongly amorphized SrPbO₃ layers (Supplementary Figure 6) show that LaLuO₃, SrPbO₃, and PbZrO₃ display a constant in-plane lattice parameter value, suggesting that the epitaxial stack is coherently strained to the LaLuO₃ buffer layer.”

3. The authors state that the GPA strain analysis was performed on the “bright field STEM image”. However, I don’t see any “bright field STEM image” in Fig. 1.

As requested by the Reviewer, we added the bright field STEM image, corresponding to the GPA strain analysis displayed in Fig. 1c-d, as Supplementary Figure 4 (Fig. R15). In the caption of Figure 1, we added:

“c-d, Out-of-plane unit-cell deformation (ϵ_{zz}) (c) and in-plane unit-cell deformation (ϵ_{xx}) (d) obtained from geometrical phase analysis of the bright field STEM image (Supplementary Figure 4).”

Fig. R15. (Supplementary Fig. 4 of the revised manuscript) Bright field STEM image on the cross-section specimen of the PbZrO_3 (30 nm) / SrPbO_3 (19 nm) / LaLuO_3 (108 nm) epitaxial stack grown on $\text{DyScO}_3(110)_o$. The corresponding geometrical phase analysis is displayed in Figure 1c-d of the manuscript.

4. Although the authors say that the SPO was sensitive to electron beam, I notice that in the strain maps of Fig. 1c and 1d, most parts of the SPO layer were still OK. Therefore, it is crucial for the authors to present magnified images that clearly show the coherent PZO/SPO interface. It is reasonable to assume that such an interface exists; it simply needs to be visually presented through magnified views.

The Reviewer is right to notice that SrPbO_3 can remain crystallized in some areas of the cross-section specimen, especially at low magnification. In Fig. R16, we provide local STEM investigations in the few areas we could find where the SrPbO_3 layer remains crystallized. We first confirm the X-ray diffraction results and demonstrate that SrPbO_3 grows with $(110)_o$ orientation and the c_o axis parallel to that of DyScO_3 , as does the LaLuO_3 orthorhombic perovskite (Fig. R16a-b). Furthermore, we clearly observe the continuity of the planes at the $\text{LaLuO}_3/\text{SrPbO}_3$ and $\text{SrPbO}_3/\text{PbZrO}_3$ interfaces, as emphasized by the HAADF-STEM image in Fig. R16c.

In the revised version of the manuscript, we added this Figure as Supplementary Figure 2 and referenced it with the sentence, page 5: “Nevertheless, local observations confirm that SrPbO_3 grows with the same $(110)_o$ orientation as the LaLuO_3 and DyScO_3 (Supplementary Figure 2).”

Fig. R16. (Supplementary Fig. 2 of the revised manuscript) STEM investigations of the SrPbO₃ layer. a, ABF-STEM image on a cross-section area with the zone axis parallel to the c_0 axis of DyScO₃. FFTs from the LaLuO₃ and SrPbO₃ layers indicate that they both have an orthorhombic structure with their c_0 axes parallel to that of the DyScO₃ substrate. b, BF-STEM image on a cross-section area with the zone axis perpendicular to the c_0 axis of DyScO₃. FFTs for LaLuO₃ and SrPbO₃ indicate that their c_0 axes lie in the film plane. c, HAADF-STEM image on a cross-section area showing the LaLuO₃/SrPbO₃/PbZrO₃ interfaces. Despite some partial amorphization at the LaLuO₃/SrPbO₃ interface, the continuity between the planes is clearly seen between the three layers with a sharp Z-contrast inversion between the SrPbO₃ and PbZrO₃, as the heavy Pb element jumps between the B-site and the A-site. Horizontal scale bars are 5 nm.

Reviewer #1 (Remarks to the Author):

The authors have addressed my concerns in the revisions, and I have no competing financial or non-financial interest in relation to this manuscript, I recommend this paper accept for publication in NC.

We thank the Reviewer for recommending the publication of our work in Nature Communications.

Reviewer #2 (Remarks to the Author):

The authors have conducted additional experiments and provided further clarification along with an improved scientific discussion. In light of these revisions, I would recommend the manuscript for publication in Nature Communications.

We thank the Reviewer for recommending the publication of our work in Nature Communications.

Reviewer #3 (Remarks to the Author):

The authors have adequately addressed the majority of the concerns from the reviewers; however, two critical issues still require further clarification in the following aspects:

We thank the Reviewer for acknowledging that we “have adequately addressed the majority of the concerns from the reviewers”. In the following, we clarify the two last issues raised by the Reviewer.

First, PbZrO_3 (PZO) is intrinsically antiferroelectric in its bulk form, the authors tend to claim that in-plane tensile strain stabilizes the antiferroelectric phase. However, this assertion lacks comparative evidence. To substantiate the strain-induced enhancement, quantitative comparisons with unstrained or bulk PZO are essential—for instance, through metrics such as polarization magnitude, ordering temperature, or the critical thickness, etc. Clarify this issue will enhance the novelty the work.

The Reviewer is right as PbZrO_3 is antiferroelectric in its bulk form, although there are some questions about the ground state of PbZrO_3 . We claim that the epitaxial strain enables the antiferroelectric state to remain down to 9 nm thicknesses, as opposed to all the recent reports showing that PbZrO_3 epitaxial films become ferroelectric or ferrielectric at the ultrathin limit (< 20 nm) [Qiao, L., Song, C., Wang, Q., Zhou, Y. & Pan, F. Polarization Evolution in Nanometer-Thick PbZrO_3 Films: Implications for Energy Storage and Pyroelectric Sensors. *ACS Appl. Nano Mater.* **5**, 6083–6088 (2022); Jiang, R.-J. *et al.* Atomic Insight into the Successive Antiferroelectric–Ferroelectric Phase Transition in Antiferroelectric Oxides. *Nano Lett.* **23**, 1522–1529 (2023); Jiang, R.-J. *et al.* A Roadmap for Ferroelectric–Antiferroelectric Phase Transition. *Nano Lett.* **24**, 11714–11721 (2024); Liu, Y. *et al.* Coexistence of ferroelectric and ferrielectric phases in ultrathin antiferroelectric PbZrO_3 thin films. *Microstructures* **4**, (2024)]. In addition, there are not, to our knowledge, any clear double hysteresis loops of polarization vs. electric field, for thicknesses of 30 nm or below, as what we show in Figure 3.

This context is summarized in our sentence, page 2: *“Initial studies showed signatures of ferroelectric phase stabilization via electric measurements^{13,14}, and recent STEM investigations revealed complex phase transitions from the classical antipolar Pbam to ferrielectric Ima2, orthorhombic or rhombohedral ferroelectric phases, as the thickness of the film is reduced to the 45–5 nm range^{15–18}.”*

This is what we compare our experimental studies with. We do not claim that the antiferroelectricity is superior to the bulk PbZrO_3 , in terms of polarization or ordering temperature. In order to clarify this point, we added the following sentence to the conclusions: *“Our results give clear evidence that the proposed epitaxy stabilizes the antiferroelectric phase against competing ferroelectric or ferrielectric phases down to thicknesses of 9 nm.”*

Second, the strain values represented in Fig. 2d, which I believe have been seriously exaggerated. A rough estimation gives the in-plane strain of +0.4% (4.181-4.163/4.163), and the out-of-plane strain of -0.2% (4.105-4.114/4.114). These values are much smaller than that presented in Figure 2d (~+1.7% and -1.4%).

We apologize here for the lack of clarity of our calculations. We believe there is some misunderstanding. We totally understand the first intuition of the Reviewer for the calculations of the epitaxial strain. In their estimate, the Reviewer consider a (001)_o orientation for PbZrO₃, with 4.163 Ang values in the film plane and 4.114 Ang value out of the plane. However, one should consider that the film is (120)_o oriented with the c_o axis of PbZrO₃ being parallel to that of DyScO₃, as clearly established by our different STEM observations. Hence, the in-plane experimental parameters calculated in Figure 2c, in green along c_o of DyScO₃ and in red perpendicular to c_o of DyScO₃, should be compared to the bulk parameters of PbZrO₃ along c_o (4.114 Ang) and perpendicular to c_o (4.163 Ang), respectively. To make the results from Figure 2d clearer, we have added a Supplementary Note 1, in which we detail our calculations of the epitaxial strain of the PbZrO₃ films. We make reference to this Supplementary Note 1 in the manuscript (page 4, page 6, and in the caption of Figure 2).

This Supplementary Note 1 is also reported below:

Supplementary Note 1 – Calculations of the epitaxial strain in PbZrO₃

Considering (120)_o PbZrO₃ on top of (110)_o SrPbO₃/LaLuO₃ with all the three orthorhombic layers having their c_o parallel, we can estimate the epitaxial strain imposed by the orthorhombic layers on PbZrO₃. Taking the bulk cell parameters of PbZrO₃, LaLuO₃, and SrPbO₃, the strain is estimated along c_o or perpendicularly to c_o as: $\frac{x_{LLO,SPO} - x_{PZO}}{x_{PZO}}$ %.

	PbZrO ₃	LaLuO ₃	SrPbO ₃
a _o (Å)	5.882	5.810	5.852
b _o (Å)	11.783	6.013	5.969
c _o (Å)	8.228	8.373	8.324
// c _o (Å)	4.114	4.187	4.162
⊥ c _o (Å)	4.163	4.181	4.180
strain PbZrO ₃ // c _o (%)		1.76	1.17
strain PbZrO ₃ ⊥ c _o (%)		0.44	0.41

Supplementary Table 1. Bulk orthorhombic unit-cells of PbZrO₃, SrPbO₃, and LaLuO₃ and estimated strain on PbZrO₃.

Hence, we clearly see from Supplementary Table 1 that the in-plane strain imposed by LaLuO₃ or SrPbO₃ on PbZrO₃ is anisotropic, with +0.4% and +1.2-1.8%, perpendicularly or along c_o, respectively. This is due to the fact that the lattice mismatch between PbZrO₃ and SrPbO₃ or LaLuO₃ is larger along c_o, than along a_o or b_o.

To calculate the epitaxial strain experimentally measured in the PbZrO₃ thin films with various thicknesses, we compared the experimental cell parameters estimated from the X-ray diffraction reciprocal space maps with the bulk parameters of PbZrO₃ (Supplementary Table 2). The evolution of these parameters with the film thickness is displayed in Figure 2c. In the last three rows of Supplementary Table 2, we report the epitaxial strain along the three axes, calculated as: $\frac{x_{film} - x_{bulk}}{x_{bulk}}$ %.

film thickness	9	20	30	40	80	200	bulk
$\perp c_0$ (Å)	4.177	4.178	4.178	4.177	4.178	4.171	4.163
// c_0 (Å)	4.187	4.186	4.184	4.184	4.184	4.179	4.114
out-of-plane (Å)	4.113	4.108	4.106	4.106	4.109	4.107	4.163
strain $\perp c_0$ (%)	0.35	0.38	0.37	0.34	0.37	0.20	
strain // c_0 (%)	1.76	1.74	1.71	1.71	1.70	1.59	
strain out-of-plane (%)	-1.18	-1.31	-1.37	-1.36	-1.30	-1.34	

Supplementary Table 2. Experimental pseudo-cubic cell parameters estimated from X-ray diffraction reciprocal space maps and calculated epitaxial strain considering (120)_o orientation.

The experimental strain values reported in this Supplementary Table 2 demonstrate that the films are under anisotropic tensile strain in the plane, resulting in an out-of-plane compression of the unit-cell. From the comparison with Supplementary Table 1, it appears that the LaLuO₃ buffer layer is imposing this amount of anisotropic tensile strain to the PbZrO₃. The thickness dependence of the experimental epitaxial strain of the PbZrO₃ films is displayed in Figure 2d.